# SWITCHCRAFT: A Programmatic Framework for Designing State-Switching Proteins

Bowen Jing [* 1]  Mihir Bafna [* 1]  Anisha Parsan [1]  Heyuan Michael Ni [2]  David Kwabi-Addo [1]
Bryan Bryson [2]  Adam Klivans [3]  Bonnie Berger [1 4]

## Abstract

Multistate mechanisms underlie many of the complex functions observed in natural proteins. The ability to rationally design multistate proteins would have transformative implications for many areas of biotechnology, yet lies beyond the capabilities of existing deep learning frameworks for protein design. To address this gap, we introduce SWITCHCRAFT, a versatile and programmatic framework for designing state-switching proteins based on backpropagation through compositional design constraints parameterized by structure prediction models. *In silico* evaluations demonstrate success on a wide range of state-switching functional primitives, from allosteric regulation of motifs to discrimination of bound ligand identities. Using these primitives, we demonstrate an *in silico* strategy for *de novo* design of fluorescent biosensors to arbitrary small molecule analytes. These results position SWITCHCRAFT at the inception of a powerful paradigm for higher-order functional protein design. Code is available at https://github.com/bjing2016/switchcraft.

## 1. Introduction

Proteins are macromolecular machines that carry out a wide range of chemical, mechanical, and computational processes, and developing methods for designing proteins with desired functions is a longstanding aim in biotechnology (Kortemme, 2024; Chu et al., 2024). Recently, generative models have driven unprecedented advancements in computational protein design. Protein language models (PLMs), trained on datasets of existing functional proteins, can generate novel sequences with similar or improved properties to existing ones (Madani et al., 2023; Hayes et al., 2025; Ruffolo et al., 2025; Lambert et al., 2025); while structure-based diffusion generative models have become widely used for the design of *de novo* binders (Watson et al., 2023; Krishna et al., 2024) and scaffolding of simple enzymatic sites (Lauko et al., 2025; Ahern et al., 2025). However, natural proteins exhibit a far richer and diverse set of functions, often featuring multistate dynamics (Grant et al., 2010)—canonical examples include motor proteins that walk along microtubules (Roberts et al., 2013), rotary mechanisms in ATP synthase (Okuno et al., 2011) and bacterial flagella (Lee et al., 2010), or information-processing polymerases that comprise the central dogma of biology. Unfortunately, neither family of technologies provides a clear path towards the design of such novel and complex functions: PLMs admit only coarse grained control via family labels or GO terms (referencing existing protein functions) rather than fine grained specifications of novel functions, while structure generators are constrained to functions described by static structures. Together, they offer only a limited sample of the full menu of functions prescribed by nature, prompting the question: how can we enable the rational design of proteins with novel complex functions?

In principle, one could train generative models given datasets pairing protein sequences or structures with detailed functional behaviors and motions, such as ligand-dependent conformational changes, state-dependent binding affinities, or mechanochemical cycles. However, no such dataset exists at scale, motivating the development of alternative frameworks beyond purely data-driven approaches.

We introduce SWITCHCRAFT, a programmatic framework for generating proteins that activate, deactivate, or switch between functional states based on the presence of a ligand effector. Our method builds upon techniques for protein design that use backpropagation on structure prediction models, which have been highly effective at designing protein and small molecule binders (Pacesa et al., 2025; Cho et al., 2025). In our framework, a generic protein design prob-

*Equal contribution [1]CSAIL, MIT [2]Dept. of Biological Engineering, MIT [3]Dept. of Computer Science, UT Austin [4]Dept. of Mathematics, MIT. Correspondence to: Bowen Jing <bjing@mit.edu>, Mihir Bafna <mihirb14@mit.edu>, Bonnie Berger <bab@mit.edu>.

*Proceedings of the 43rd International Conference on Machine Learning*, Seoul, South Korea. PMLR 306, 2026. Copyright 2026 by the author(s).

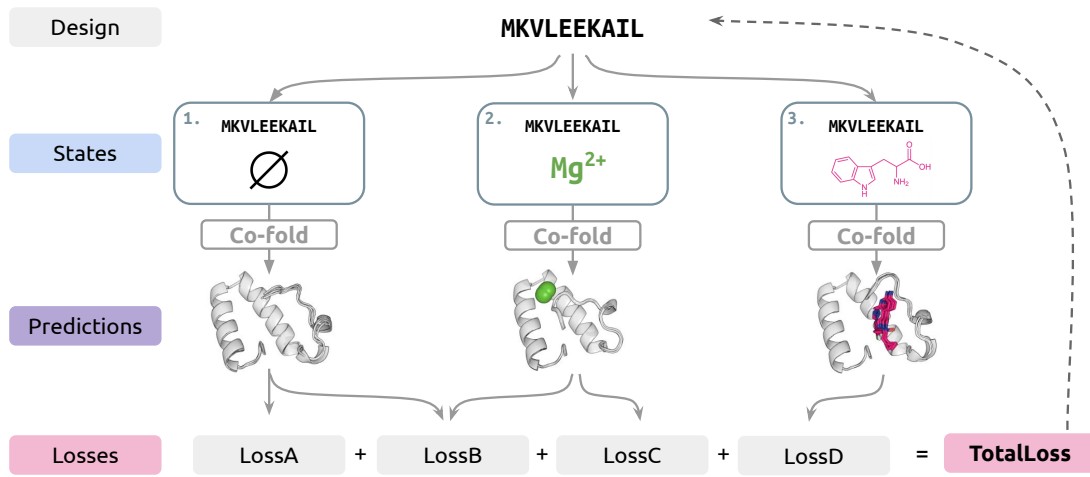

*Figure 1.* **SWITCHCRAFT design of multistate proteins.** Designs are specified by defining multiple states and corresponding structural constraints (losses). At each iteration, a single sequence is co-folded with each state's molecular context and state-specific losses are evaluated. The sequence is then optimized by gradient descent to satisfy all states jointly.

lem is specified via an arbitrary number of *states*, each of which is described via an arbitrary number of *constraints* that can be evaluated using the output of the structure prediction model. The task of designing a multistate functional protein is formalized as that of finding a sequence that optimally satisfies all these constraints (Figure 1). As such, our method is general and can be applied to any functions described using a set of structural properties partitioned across multiple states. To our knowledge, this provides the first general-purpose computational framework for multistate protein design.

Our experiments explore several different variations on this theme, as illustrated in Figure 2. For the simplest case of single ligands activating (positive allostery) or deactivating (negative allostery) a structural motif, we systematically use a diverse set of ligands, including small molecules, metal ions, and DNA, to modulate 11 of the 24 motifs from the well-known RFDiffusion motif scaffolding benchmark (Wang et al., 2022; Watson et al., 2023). We then demonstrate individual designs for more complex design specifications: proteins that switch between two motifs with a ligand effector (motif switching); that bind a target protein only in the presence of a ligand (induced binding); that exhibit a large conformational shift upon change to a bound ligand (ligand modification); and that switch between three distinct conformations depending on which of two ligands is bound (ligand discrimination). We obtain these designs with relatively little tuning and filtering, even for highly compositional design specifications. Finally, we demonstrate a computational workflow for designing cpGFP-based fluorescent biosensors for small molecule analytes, a design setting of scientific, practical, and commercial value.

## 2. Background

**Generative modeling for protein design.** Techniques for designing proteins with generative models can be broadly categorized as sequence generative models, often called protein language models (PLMs), and structure design methods. PLMs learn the distribution of hundreds of millions of natural protein sequences, conditioned on attributes such as the protein family (e.g. Pfam) or Gene Ontology terms imputed from homology (Ferruz & Höcker, 2022; Jing et al., 2025). When prompted with such attributes, PLMs have generated functional proteins such as lysozymes (Madani et al., 2023), tryptophan synthases (Lambert et al., 2025), green fluorescent proteins (Hayes et al., 2025) and even CRISPR nucleases (Ruffolo et al., 2025). Clearly, PLMs are capable of understanding complex functions. However, because the conditioning tag (and associated protein sequences) must be available at training time, this kind of conditioning does not permit the specification of *novel* functions for design.

On the other hand, structure design methods are developed to generate structures that are plausible binding partners for a target or scaffolds for a functional motif. These are either diffusion or flow-matching models which learn the distribution of protein structures in the PDB (Watson et al., 2023; Bose et al., 2023; Geffner et al., 2025a;b), or methods that backpropagate through structure prediction models (Pacesa et al., 2025; Cho et al., 2025; Mille-Fragoso et al., 2025). Although such techniques have attracted significant attention for their value in drug discovery and have demonstrated *de novo* design of simple enzymes (Ahern et al., 2025; Butcher et al., 2025), they are fundamentally limited to generating individual *structures* and therefore cannot accommodate the specification of complex functions requiring multiple states.

**Multistate protein design.** Prior approaches for multistate protein design have relied significantly on manual design choices, with limited adoption of deep learning frameworks. Early *de novo* efforts have primarily focused on constructing rigid domains that undergo hinge-like motions in response to ligand binding or behavior reminiscent to that of natural systems like hemoglobin (Praetorius et al., 2023; Pillai et al., 2024). Related strategies link together naturally occurring domains using designed connectors to enable inter-domain communication and allosteric coupling (Pirro et al., 2020). Other approaches begin from existing protein scaffolds that bind a target of interest and iteratively modify it through experimental screening and computational refinement to achieve multistate behavior (Broerman et al., 2025; Guo et al., 2025). While successful in individual cases, these strategies require significant domain expertise and are difficult to generalize to arbitrarily complex multistate design specifications.

More recent methods have sought greater automation but still fall short of full *de novo* multistate design. Tied ProteinMPNN (Dauparas et al., 2022) and DynamicMPNN (Abrudan et al., 2025) sample sequences conditioned on multiple backbones, but do not provide those backbones. ProDiT (Jing et al., 2025) and ProteinGenerator (Lisanza et al., 2025) aim to generate protein sequences corresponding to multiple backbones by tying together generation trajectories with a sequence track. However, neither model accepts ligand atoms as input, providing no control over the putative states. Consequently, these methods do not provide a general framework for designing multistate mechanisms.

## 3. Method

In the SWITCHCRAFT design protocol, we frame multistate protein design as sequence optimization under an objective function that describes consistency with the desired multistate behavior, where the objective function is parameterized with structure prediction models. As such, the design process consists of two stages: **design specification** (Section 3.1) and **design optimization** (Section 3.2). In the specification stage, we construct a global constraint function that evaluates a partially-designed sequence representation $\mathbf{z} \in \mathbb{R}^{20 \times L}$ under several instantiations of Boltz-1 (Wohlwend et al., 2025), corresponding to multiple states, and returns a scalar that represents the deviation from the target design criteria. In the optimization stage, we gradually refine $\mathbf{z}$ towards a one-hot representation of a protein sequence using gradients from the global constraint function and a multi-stage optimization protocol adapted from BoltzDesign-1 (Cho et al., 2025). This framing is reminiscent of the model definition and training loop of a deep learning model; as such, we call our constraint functions *loss* functions.

### 3.1. Design specification

A multistate design specification consists of states $s = 1, \ldots N_{\text{states}}$, each associated with a *folding context* $\mathcal{C}_s$, which is a (potentially empty) set of fixed molecules; a set of loss functions $\mathcal{L}_n : \mathbb{R}^{20 \times L} \to \mathbb{R}$ which expresses the desired behavior of the target sequence under the folding contexts; a design mask $\mathbf{m} \in \{0, 1\}^L$ denoting which residue positions are to be designed; and an optional motif sequence $\mathbf{s} \in [1, 20]^L$ for the residue identities that are not designed. Each loss depends on a subset of the states and prototypically has the functional form $\mathcal{L}_n(\mathbf{z}) = \mathcal{L}'_n(\text{Boltz-1}(\mathbf{z}, \mathcal{C}_s))$, i.e., the loss function operates over the outputs of Boltz-1, although losses that use other models or only operate on sequence properties can be conceived. Here, we define and use the following losses:

**Motif loss.** Given a motif with residue indices $m \subset [1, L]$ and corresponding $C\beta$ positions $\mathbf{r}$ ($C\alpha$ for glycine), we optimize for scaffolding the motif via

$$\mathcal{L}_{\text{motif}} = \sum_{\substack{i,j \in m \\ i \neq j}} \sum_k \frac{p_{ijk}}{|m|(|m| - 1)} \left( d_k - \|\mathbf{r}_i - \mathbf{r}_j\| \right)^2 \tag{1}$$

where $p_{ijk}$ is the distogram output of Boltz-1 for the $i, j$ residue pair and $k$th bin and $d_k$ is the midpoint distance of the $k$th bin. This averages the expected squared error of each pairwise distance constrained by the motif. Correspondingly, we optimize for *not* scaffolding the motif via the **anti-motif loss** $\mathcal{L}_{\text{anti-motif}} = -0.5\mathcal{L}_{\text{motif}}$.

**Binding loss.** Given a ligand, we optimize the design to bind the ligand via a loss function adapted from BoltzDesign1 (Cho et al., 2025):

$$\mathcal{L}_{\text{binding}}(\mathbf{z}) = \frac{1}{2c} \sum \text{mink}_j^{(k=c)} \text{mink}_i^{(k=2)} H_{<20\text{Å}}(D_{ij}) \tag{2}$$

where $D_{ij}$ is the distance between the $i$th protein $C\beta$ position and the $j$th ligand and its distribution is taken from the distogram output. Abusing notation, the thresholded entropy $H_{<k}$ is defined as:

$$H_{<k}(D_{ij}) = -\mathbb{E}[\log P(D_{ij}) | D_{ij} < k] \tag{3}$$

$$= -\mathbb{E}[\log P(D_{ij} \mid D_{ij} < k) \mid D_{ij} < k] - \log P(D_{ij} < k) \tag{4}$$

That is, it promotes confident contacts via low entropy in the renormalized distance distribution within the cutoff and a high contact log-probability. The binding loss aggregates the confidence of the top two contacts per ligand token for the top $c$ tokens, where $c$ varies from 8 to 12 over the course of the optimization. Correspondingly, we optimize for *not* binding the ligand via the **anti-binding loss** $\mathcal{L}_{\text{anti-binding}} = -0.5\mathcal{L}_{\text{binding}}$.

**Conformational change loss.** To optimize for a conformational change between two states with folding contexts $\mathcal{C}_1, \mathcal{C}_2$ we define the loss

$$\mathcal{L}_{\text{conf-change}}(\mathbf{z}; \mathcal{C}_1, \mathcal{C}_2) = -\frac{1}{L}\sum_{i=1}^{L}\max_{j\in[1,L]}\text{JSD}(D_{ij}^{(1)} \,\|\, D_{ij}^{(2)}) \tag{5}$$

where $D_{ij}^{(1)}, D_{ij}^{(2)}$ are the distograms for residue pair $i, j$ from the Boltz-1 outputs on folding context $\mathcal{C}_1, \mathcal{C}_2$, respectively, and JSD is the Jensen-Shannon divergence. Minimizing the JSD pushes the distance distributions to differ between states; the conformational change loss maximizes this change for the contact with the largest sensitivity for each residue.

**Contact loss.** Finally, to preserve that each protein state is well-predicted, we retain the intra-design contact loss from BoltzDesign1 (Cho et al., 2025):

$$\mathcal{L}_{\text{contact}}(\mathbf{z}) = \frac{1}{L}\sum_{j=1}^{L}\min_{i:|i-j|\geq 9} H_{<14\text{\AA}}(D_{ij}) \tag{6}$$

This maximizes the confidence of the top long-distance contact per residue position. We implicitly include this loss for all states in all design settings.

The global loss function is evaluated by first passing the designed sequence $\mathbf{z}$ along with all folding contexts through Boltz-1, and summing all constituent loss terms using the cached outputs. Note that while each folding context can be thought of as a state, the molecules in each folding context are not necessarily meant to interact, as the anti-binding loss will explicitly penalize specified interactions.

### 3.2. Design optimization

Given a fixed design specification, our optimization seeks to design a sequence that adheres to the specified constraints by iteratively updating a logit-parameterized sequence representation $\mathbf{z} \in \mathbb{R}^{20\times L}$ using gradients of the global loss. Optimization proceeds for 240 steps using a four stage schedule adapted from BoltzDesign1 (Cho et al., 2025) as follows:

Each column of $\mathbf{z}$ corresponds to logits over amino acid identities at a residue position. The soft categorical representation is given by

$$\mathbf{z}_{\text{soft}} = \text{softmax}(\mathbf{z}/\tau), \quad \mathbf{z}_{\text{soft},:,i} \in \Delta^{20} \tag{7}$$

while the corresponding discrete sequence is given by the hard projection

$$\mathbf{z}_{\text{hard}} = \text{onehot}(\text{argmax}\,\mathbf{z}) \in \{0,1\}^{20\times L} \tag{8}$$

Because the argmax is non-differentiable, a straight-through-estimator (STE) is utilized, admitting loss evaluation on

the hard sequence while allowing for gradient propagation through the soft relaxation:

$$\mathbf{z}_{\text{st}} = (\mathbf{z}_{\text{hard}} - \mathbf{z}_{\text{soft}})\Big|_{\nabla=0} + \mathbf{z}_{\text{soft}} \tag{9}$$

By construction, $\mathbf{z}_{\text{st}} \equiv \mathbf{z}_{\text{hard}}$ during loss evaluation, while $\nabla_z \mathbf{z}_{\text{st}} = \nabla_z \mathbf{z}_{\text{soft}}$, providing a continuous surrogate.

The sequence representation passed into `Boltz-1` is constructed as a convex combination of the straight-through, soft, and logit representations,

$$\mathbf{z}_{\text{pseudo}} = \beta\mathbf{z}_{\text{hard}} + (1-\beta)(\gamma\,\mathbf{z}_{\text{soft}} + (1-\gamma)\mathbf{z}) \tag{10}$$

with hyperparameters $\beta, \gamma$, and temperature $\tau$ set to modulate the emphasis between continuous and discrete, enabling early optimization stages to favor smooth exploration and later stages to anneal towards discrete residue identities.

Note that if a motif mask $\mathbf{m}$ is present, the motif residue types would be placed in $\mathbf{z}_{\text{pseudo}}$ as one-hot encodings before the forward pass of Boltz-1. In this case, only residue positions $i$ with $\mathbf{m}[i] = 1$ are optimized, with the remaining positions fixed to the motif sequence $\mathbf{s}$ for all loss evaluations. For motif placement specification, we adopt the sampling protocol introduced by (Lin et al., 2024). When scaffolding multiple motifs, we first apply Algorithm 2 to merge motif constraints before sampling.

Algorithm 1 summarizes a single optimization step, including the construction of the pseudo-sequence representation, motif clamping when present, and the corresponding gradient update under the current schedule parameters.

---

**Algorithm 1** Single step sequence optimization

---

$\mathbf{z}_{\text{soft}} \leftarrow \text{softmax}(\mathbf{z}/\tau)$
$\mathbf{z}_{\text{hard}} \leftarrow \text{onehot}(\text{argmax}(\mathbf{z}))$
$\mathbf{z}_{\text{hard}} \leftarrow (\mathbf{z}_{\text{hard}} - \mathbf{z}_{\text{soft}}).\text{detach}() + \mathbf{z}_{\text{soft}}$
$\mathbf{z}_{\text{pseudo}} \leftarrow \beta\mathbf{z}_{\text{hard}} + (1-\beta)\gamma\mathbf{z}_{\text{soft}} + (1-\beta)(1-\gamma)\mathbf{z}$
$\mathbf{z}_{\text{pseudo}}[\mathbf{m} == 0] \leftarrow \text{onehot}(\mathbf{s})$
$\mathbf{z} \leftarrow \mathbf{z} - \alpha\nabla_{\mathbf{z}}\mathcal{L}(\texttt{Boltz-1}(\mathbf{z}_{\text{pseudo}}))$

---

The overall optimization protocol initializes $\mathbf{z}$ from a Gumbel-softmax distribution and proceeds in four stages:

**Stage 1.** 30 steps with $\alpha = 0.2$, $\beta = 0$, $\gamma = 1$, and $\tau = 0.5$, followed by $\mathbf{z} \leftarrow \mathbf{z}_{\text{pseudo}}$

**Stage 2.** 100 steps with $\alpha = 0.1$, $\beta = 0$, $\gamma$ interpolating from 0 to 1, and $\tau = 0.5$, followed by $\mathbf{z} \leftarrow 2\mathbf{z}$

**Stage 3.** 100 steps with $\alpha = 0.1$, $\beta = 0$, $\gamma = 1$, and $\tau$ interpolating from 0.5 to 0.005

**Stage 4.** 10 steps with $\alpha = 0.1$, $\beta = 1$, $\gamma = 1$, and $\tau = 0.005$

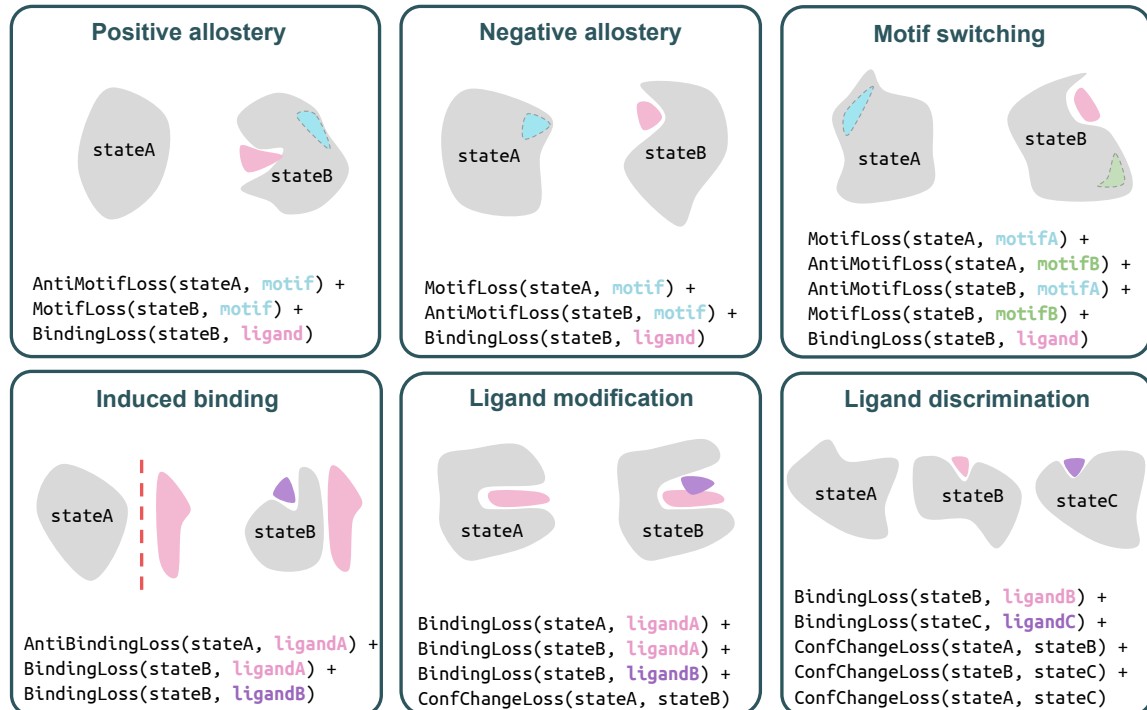

*Figure 2.* **Versatile and programmatic multistate design capabilities enabled by SWITCHCRAFT.** In each setting, we seek to design a protein (grey) to adopt the multiple states illustrated. Motifs to be scaffolded are colored blue or green, while potential binding partners (ligands) are shown in pink or purple. All specifications also include a `ContactLoss` term for each distinct protein state.

## 4. Experiments

We first validate SWITCHCRAFT on six design settings of increasing degrees of complexity, which we view as fundamental primitive tasks of multistate protein design. Each of the settings is specified in terms of their constituent losses in Figure 2. We then apply SWITCHCRAFT to a more complex application—computational biosensor design (Sec. 4.6)—by reducing it to the design of a ligand-responsive conformational switch. For all settings, we run a large number of independent optimization trajectories, predict five structures with `Boltz-1` for the final designed sequence in each state, and evaluate designs based on several quality criteria defined in Appendix A.2. Further details, statistics, and results for all experiments, as well as additional controls and ablations, can be found in Appendix B.

### 4.1. Positive and negative allostery

In the simplest case of multistate design, we seek to scaffold a backbone motif in the presence or absence of a ligand such that ligand unbinding or binding significantly modulates motif integrity. For each of five ligands—small molecules OQO and flavin adenine dinucleotide (FAD); metal ions $Zn^{2+}$ and $Mg^{2+}$; and dsDNA with sequence GAATTC—we systematically designed scaffolds with positive and negative allostery for each of the 24 motifs from the RFDiffusion benchmark

(Watson et al., 2023). We designed 100 sequences for each problem, specification, and ligand, resulting in 11 motifs with at least one successful design (Figure 6). Successful designs are defined by a clear, state-dependent change in motif scaffold quality: in *positive allostery*, motifs are disrupted in the unbound state and confidently scaffolded upon ligand binding, while the inverse behavior is required for *negative allostery*. In both cases, we additionally require low intra-state variability in motif RMSD to ensure confident structural predictions and a substantial difference in mean motif RMSD ($> 1$Å) between states to confirm a meaningful conformational response. Examples of successful designs visualized in Figure 3 exhibit large disruptions of the scaffolded motif, often exceeding 5 Å, associated with pronounced conformational changes and, in some cases, fold switching (notably for dsDNA). These changes are consistently predicted across diffusion samples, with low intra-state RMSDs ($\leq 2$Å) and low variance in motif RMSDs in each state (Figures 7, 8, 9). Detailed filtering criteria and additional analyses are provided in Appendix B.

With the wide array of motif and ligand types evaluated, we aim to establish this positive/negative allostery design task as a benchmark for computational multistate design methods (Figure 6). Although our results are encouraging, success rates remain low in absolute terms and provide ample room for further methodological improvement.

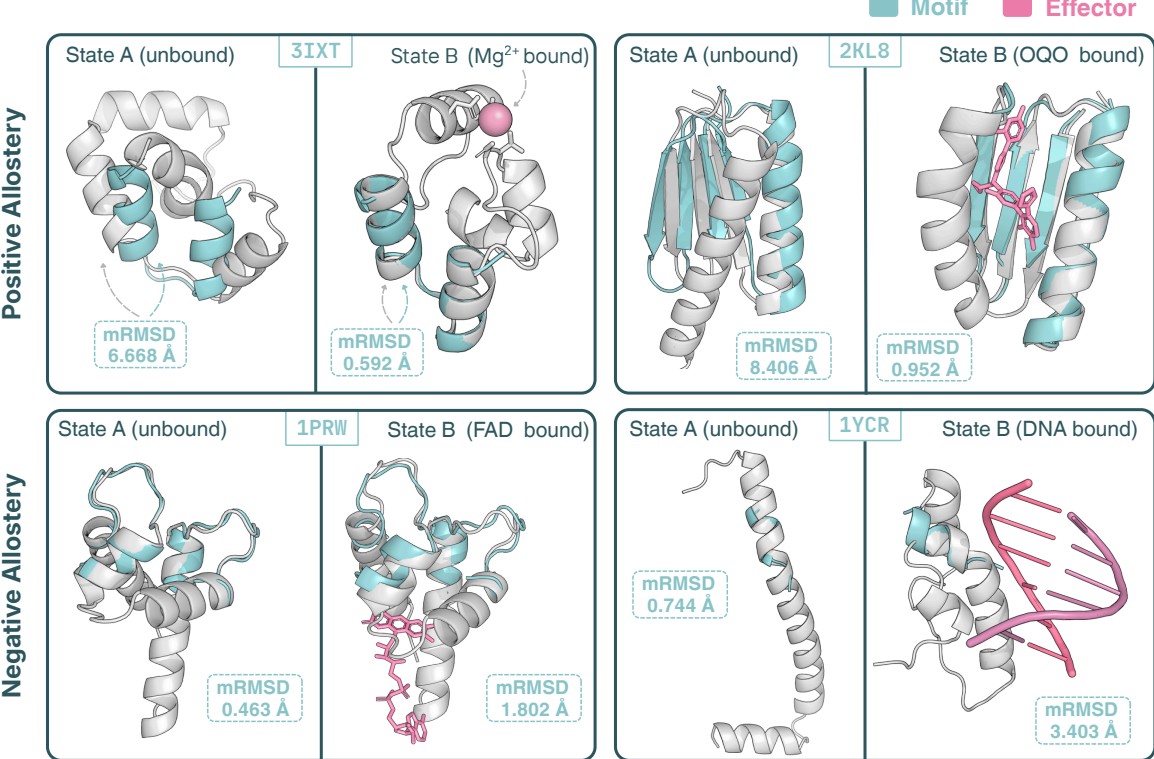

*Figure 3.* **Selected examples of SWITCHCRAFT designs for positive and negative allostery** across small-molecule, metal-ion, and dsDNA ligands, with $C\alpha$ motif RMSD. Motifs are shown in teal, ligands in pink, and designs in gray. Further analysis in Appendix B.

## 4.2. Motif switching

Next, we sought to design proteins that toggle between two functional motifs upon ligand binding. Such functionality could be useful in designing dual-purpose proteins that are responsive to environmental inputs, such as pairs of enzymatic or regulatory functions (Ha & Loh, 2012). We devised a means of merging motif scaffold specifications (Algorithm 2) and constructed design specifications to toggle between motifs 3IXT and 1YCR, which exhibited the highest success rates in positive and negative allostery. Indeed, motif switching can be viewed as positive allostery for the first motif and negative allostery for the second, thus we evaluate designs using the same cutoff heuristics as in the previous section. Out of generated 100 designs with OQO as the effector, 3 exhibited confidently predicted scaffolding of both motifs in their respective states and disruption of each motif in the converse state (Figure 10). An example is shown in Figure 4, top left and Figure 11. Notably, many other designs satisfied three of the four desired constraints, successfully modulating one motif across states while the second motif remained scaffolded, falling just short of full motif switching.

## 4.3. Ligand modification

Many natural proteins bind ligands that possess conformational or electronic properties, such as affinity for dissolved gases (Gondim et al., 2022) or light sensitivity (Conrad et al., 2014), which are otherwise difficult to accomplish with amino acid chemistry. We sought to design proteins that could similarly leverage bound ligands to augment protein functionality.

As a prototypical example, we constructed a design specification for a heme-binding protein that would exhibit a conformational change upon addition of molecular oxygen. Out of 558 designs, 10 proteins exhibited such changes (Figure 12); in our selected example, a coordinating histidine is displaced by the oxygen, inducing a 3.8 Å conformational change (Figure 4, top right; Figure 13). These designs satisfy a stringent specification requiring both stable heme coordination and ligand-induced rearrangement of the coordination environment. This conformational change is reminiscent of cooperativity in hemoglobin, which features a similar iron coordination site with two histidines (Ahmed et al., 2020). Interestingly, some examples (Figure 14) exhibited large conformational changes, but involved rearrangements that appeared implausible without unbinding and rebinding. This suggests future value in *kinetics*-based constraints to ensure plausible state transitions.

*Figure 4.* **Case studies for motif switching, ligand modification, induced binding, and ligand discrimination design** specifications. C$\alpha$ motif-, intra-, and cross-RMSDs, as well as iPTM scores, are shown when applicable. Motifs are shown in blue or green, binding partners in pink or purple, and designed proteins in gray. Further analysis and results in Appendix B.

## 4.4. Induced binding

Many methods have been developed for designing high affinity binders; however, comparatively little work has focused on modulating binding itself—such as turning interactions on or off, or tuning binding strength in response to an effector, for example across a metabolite concentration gradient. Indeed, control over a protein's interactions with other molecular species underlies many biological functions; for example, protein kinase A releases catalytic subunits only upon cAMP binding (Taskén & Aandahl, 2004), and Ca$^{2+}$ interacting with calmodulin exposes hydrophobic interfaces (Zhang et al., 2012). We explore this class of functional control by designing proteins to engage a partner only in the presence of an effector which stabilizes or creates an interaction interface.

Specifically, we sought to design a 50 AA protein that interacts with a 16 amino acid fragment of Top7 (Kuhlman et al., 2003) only in the presence of Ca$^{2+}$. We use ipTM as a proxy for protein-peptide binding. Out of 940 designs, 8 showed significantly higher ipTM in the presence of calcium with confidently predicted structures in both states (Figure 15). Our visualized design (Figure 4, bottom left; Figure 16) undergoes a 12.50 Å conformational change upon

Ca$^{2+}$ binding, repositioning residues to form an interaction interface with the Top7 fragment.

## 4.5. Ligand discrimination

Many proteins adopt more than two distinct ligand-dependent states. Enzymes often must catalyze a series of reactions each accompanied by conformational changes, and G protein-coupled receptors (GPCRs) engage in distinct signaling pathways depending on the bound ligand (Rankovic et al., 2016). We explore this ability of our protocol to target such specifications by designing a 50 residue miniprotein to bind the ligands OQO and Ca$^{2+}$ in separate states, each distinct from the unbound state.

From 12 out of 465 successful designs (Figure 17), our visualized design possesses a key loop that adopts three distinct conformations (Figure 4, bottom right; Figure 18): forming a Glu/Arg salt bridge in the unbound state, rearranging upon OQO binding to create a hydrophobic pocket in the second state, and then undergoing an even more substantial conformational shift to accommodate a calcium coordination site in the third state. The designed loop displays deviations of at least 1.48 Å across any two states, and is also confidently predicted in all three states with intraRMSDs

**A. Schematic**

**B. Real Example**

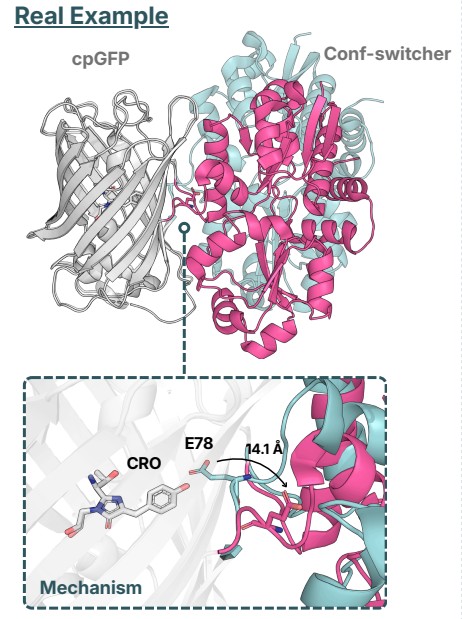

**C. SwitchCraft Design**

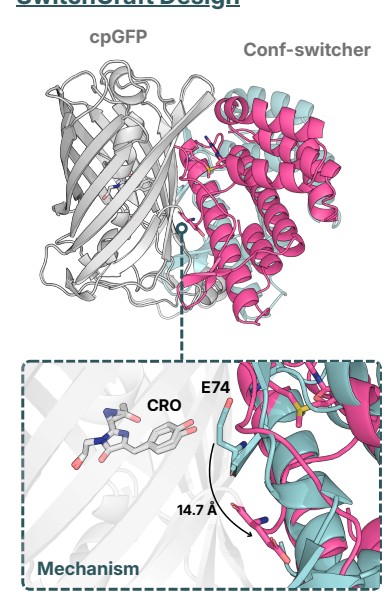

*Figure 5.* **SWITCHCRAFT-based computational workflow for biosensor design.** (a) Schematic of biosensor design: circularly-permuted green fluorescent protein (cpGFP) inserted into ligand-specific conformation switching protein allowing for modulation of chromophore (CRO) contacts and fluorescence. (b) Real example of designed nicotinic biosensor (PDB: `7s7u`, `7s7v`) exhibiting chromophore un-quenching. (c) SWITCHCRAFT designed SAM biosensor highlighting the same mechanism, where upon ligand binding, `Glu74` shifts away from the chromophore enabling fluorescence.

consistently less than $1.0\text{Å}$. The successful targeting of such multi-ligand specifications hints towards the rational design of multistep enzymes with many intermediate species.

### 4.6. Workflow for biosensor design

As a stress test of the multistate design abilities of SWITCHCRAFT, we develop a pipeline for the design of *de novo* fluorescent biosensors for arbitrary small molecules. Designing such biosensors has been a long-standing challenge for their relevance in probing intracellular signaling pathways and monitoring drug responses over time. Most biosensor designs are fusion proteins comprising (1) a fluorescent reporter domain such as circularly permuted GFP (cpGFP) and (2) a ligand-responsive conformational switch which modulates the fluorescence of the reporter (see Fig. 5a). In successful designs, the cpGFP is inserted into the conformational switch such that ligand binding perturbs the chromophore environment (Sec. A.4). Sensors for calcium (Nakai et al., 2001), maltose (Marvin et al., 2011), glutamate (Marvin et al., 2013), zinc (Qin et al., 2016), nicotine (Nichols et al., 2022) among many other molecules, have been developed in this manner.

However, this design workflow is predicated on the availability of a natural conformational switch for the desired small molecule. Although biosensors for new ligands can sometimes be obtained via directed evolution campaigns, this process can be time-consuming. Here,

we use SWITCHCRAFT to design conformation switchers that are responsive to three small molecule ligands—S-adenosylmethionine (SAM), cyclic guanosine monophosphate (cGMP), and adenosine triphosphate (ATP). To do so, our design specification comprises a `ContactLoss` in both apo and holo states to ensure structural confidence, a `BindingLoss` in the holo state between the protein and ligand, and a `ConfChangeLoss` across states to ensure significant structural deviation. Across the three ligands, we generated 13,858 total designs with lengths ranging from 150 to 200. Of these, 89 satisfied a stringent set of criteria: effector iPTM $> 0.8$, intraRMSD $\leq 1.5\text{Å}$ and average pLDDT $> 0.8$ in both states, crossRMSD $> 3\text{Å}$, and compact folds (radius of gyration $\leq 22\text{Å}$).

Next, to emulate the setting of assembling a biosensor with these conformation switchers, we sought to replicate the mechanism of action of the nicotine biosensor in the cpGFP-confswitcher fusion (Fig. 5b). In particular, the apo state places a linker glutamic acid (Glu78) in direct contact with the chromophore, which quenches fluorescence, while ligand binding induces a conformational change that tugs this residue away from the chromophore (14.1 Å shift), enabling fluorescence. For each conf-switcher, we selected cpGFP insertion sites at residues exhibiting the largest backbone dihedral angle changes across the apo and holo states, following Marvin et al. (2011), while enforcing spatial diversity between sites (see Alg. 3). We then inserted cpGFP at

these sites, co-folded the resulting constructs, and screened for significant changes in chromophore contacts to identify the designs with proper chromophore modulation. Under this criterion, 44 designs passed. Notably, we identify a SWITCHCRAFT-designed SAM biosensor that exhibits the same chromophore "unquenching" mechanism (see Fig. 5c, 14.7 Å shift in Glu74) as the nicotine biosensor.

## 5. Conclusion

In this work, we developed the first systematic framework for multistate protein design, SWITCHCRAFT, which allows us to more broadly sample the menu of protein functions enjoyed by nature. By refining a sequence with multiple *implicit* structural constraints, we circumvent the conceptual bottleneck of prevailing single-structure design paradigms. We validated our method *in silico* on six functional multistate primitives and demonstrated practical implications in a biosensor design case study. With the additional development of constraints based on *atomic* motifs, along with creative multistate specifications for complex functions, SWITCHCRAFT is well poised to mature into a powerful framework for rational design of novel protein functions. Future work will focus on experimental validation of designs for the multistate primitives (preliminary experiments for induced binding in Appendix B.7) and for more complex specifications. Ultimately, we envision multistate design as a language with which we can emulate the wide and dizzying array of functions found in nature—motor proteins, rotary mechanisms, or even the information-processing polymerases that lie at the heart of biology.

## Acknowledgments

We thank Daniel J. Diaz, Y. Jessie Zhang, Rohith Krishna, Anand Muthusamy, and Varun Ullanat for helpful feedback and discussions. This work was supported by the National Institute of General Medical Sciences of the National Institutes of Health under award 1R35GM141861 (to B.B.), the NSF AI Institute for Foundations of Machine Learning (IFML), and the UT-Austin Center for Generative AI.

## Impact Statement

SwitchCraft aims to expand protein design beyond static structures toward programmable, state-dependent function. By enabling rational design of proteins that switch in response to specific molecular inputs, this framework could eventually support new biosensors, controllable therapeutics, and synthetic biology tools.

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

# A. Additional Method Details

## A.1. Design optimization

To design a sequence for a given design specification, we iteratively evaluate the loss and use its gradients to update the sequence representation. A single step of sequence optimization is shown in Algorithm 1. Note that if a motif mask is present, the motif residue types would be placed in $\mathbf{z}_{\text{pseudo}}$ as one-hot encodings before the forward pass of Boltz-1. The overall optimization protocol initializes $\mathbf{z}$ from a Gumbel-softmax distribution and proceeds in four stages:

**Stage 1.** 30 steps with $\alpha = 0.2$, $\beta = 0$, $\gamma = 1$, and $\tau = 0.5$, followed by $\mathbf{z} \leftarrow \mathbf{z}_{\text{pseudo}}$

**Stage 2.** 100 steps with $\alpha = 0.1$, $\beta = 0$, $\gamma$ interpolating from 0 to 1, and $\tau = 0.5$, followed by $\mathbf{z} \leftarrow 2\mathbf{z}$

**Stage 3.** 100 steps with $\alpha = 0.1$, $\beta = 0$, $\gamma = 1$, and $\tau$ interpolating from 0.5 to 0.005

**Stage 4.** 10 steps with $\alpha = 0.1$, $\beta = 1$, $\gamma = 1$, and $\tau = 0.005$

After optimization, we take the argmax of all non-motif residue positions.

## A.2. Evaluation criteria.

*Motif RMSD* (mRMSD) measures the alignment error of scaffolded motifs relative to their target conformations; mRMSD $\leq 1$Å typically indicates accurate placement, with low variability across predicting structures (std $\leq 0.5$Å) serving as a proxy for confidence. *IntraRMSD* measures structural consistency by averaging pairwise RMSDs among predicted structures within the same state, while *CrossRMSD* averages RMSDs across different states to quantify large structural deviations. IntraRMSD $\leq 2$Å generally indicate confident conformations, while crossRMSD $> 3$Å indicate meaningful structural shifts, again with low standard deviations. Finally, iPTM is used to assess confidence in ligand or interface binding when applicable, with iPTM $> 0.6$ serving as a standard threshold.

## A.3. Motif specifications

For each of the 24 RFDiffusion (Watson et al., 2023) motifs, we follow the same protocol of sampling motif specifications from (Lin et al., 2024). Each motif specification consists of (1) an ordered list of segments, where each segment is either a *motif segment*, which is a contiguous set of motif residues, or a *scaffold segment*, which is described by a maximum and minimum length, and (2) a minimum and maximum global length. This specification allows us to generate multiple valid design problems by randomly sampling lengths of scaffold segments within their allowable ranges while preserving the global length constraints and the ordering of the motif. We use the Genie2 (Lin et al., 2024) algorithm for this procedure.

When scaffolding proteins that must accommodate *multiple* motif specifications simultaneously, we first merge them into a single unified motif specification. This merged specification can then be sampled using the Genie2 algorithm in the same way as a single-motif design. The merging procedure is outlined in Algorithm 2.

---

**Algorithm 2** Motif Spec Merging

    **Input:** Motif specifications $M_1, M_2, \ldots, M_k$
    Initialize $S \leftarrow M_1$
    **for** $i = 2$ to $k$ **do**
      Let $T \leftarrow$ last scaffold segment of $S$
      Let $L \leftarrow$ first scaffold segment of $M_i$
      Compute new scaffold segment to merge $S$ and $M_i$:
        min_len $\leftarrow \max(T.\text{min\_length}, L.\text{min\_length})$
        max_len $\leftarrow \min(T.\text{max\_length}, L.\text{max\_length})$
        If min_len $>$ max_len then set max_len $\leftarrow \max(T.\text{max\_length}, L.\text{max\_length})$
        Define new scaffold segment Pad $\leftarrow$ {type=scaffold, min_length, max_length}
      Concatenate segments:
        $S.\text{segments} \leftarrow S.\text{segments}[:-1] \parallel \text{Pad} \parallel M_i.\text{segments}[1:]$
      Recompute global min/max total lengths
    **end for**
    **return** merged motif specification $S$

---

### A.4. Biosensor design

#### A.4.1. ADDITIONAL BACKGROUND

Many genetically encoded fluorescent biosensors use a circularly permuted GFP (cpGFP) as the reporter domain. In cpGFP, the native N and C termini of GFP are reconnected, and new termini are introduced closer to the chromophore. This rearrangement renders the chromophore environment sensitive to local structural perturbations, enabling fluorescence modulation by conformational changes. cpGFP is then inserted into residue regions with high dihedral angle changes (Marvin et al., 2011) (see Alg. 3) and short linker sequences.

#### A.4.2. DESIGN CLARIFICATIONS

To identify candidate regions for cpGFP insertion into the conformation switcher, we utilize a simple heuristic based on conformational flexibility. For effective biosensor design we need the conformation switcher to "tug" contacts away from the chromophore, thus we select residues exhibiting large backbone dihedral changes between the apo and holo states. To avoid redundant insertions, we additionally enforce a minimum sequence separation between selected sites. Algorithm 3 summarizes this procedure.

---

**Algorithm 3** Confswitcher Insertion Site Selection

---

**Input:** Backbone dihedral changes $\Delta \in \mathbb{R}^{N \times L}$
**Parameters:** min change $\delta_{\min}$, min separation $d_{\min}$, num sites $k$
Compute per-residue median change $m_i \leftarrow \operatorname{median}(\Delta_{:,i})$
Identify candidate sites $\mathcal{C} \leftarrow \{i \mid m_i \geq \delta_{\min}\}$
Sort $\mathcal{C}$ by decreasing $m_i$
Initialize selected sites $\mathcal{S} \leftarrow \emptyset$
**for** each $i \in \mathcal{C}$ **do**
  **if** $|i - j| \geq d_{\min}$ for all $j \in \mathcal{S}$ **then**
    $\mathcal{S} \leftarrow \mathcal{S} \cup \{i\}$
  **end if**
  **if** $|\mathcal{S}| = k$ **then**
    **break**
  **end if**
**end for**
**return** insertion sites $\mathcal{S}$

---

# B. Additional Results

## B.1. Positive and negative allostery

For systematic analysis of SWITCHCRAFT's positive/negative allostery capabilities, we designed sequence scaffolds for each of the 24 motifs from the RFDiffusion benchmark (Watson et al., 2023) across 5 ligands—small molecules OQO and flavin adenine dinucleotide (FAD); metal ions $Zn^{2+}$ and $Mg^{2+}$; and dsDNA with sequence GAATTC. We generated 100 sequences for each motif problem, specification, and ligand, resulting in 11 motifs with at least one success (Figure 6).

To quantify success, we applied the following filtering criteria. Each design yields 5 diffusion sampled structures per state (in this case two: A/B).

- **Positive allostery** (ligand promotes motif functionality): designs were required to have average motifRMSD $> 1.0$ Å in the unbound state A (improperly scaffolded) and $\leq 1.0$ Å in the bound state B.

- **Negative allostery** (ligand disrupts motif functionality): the inverse criteria was applied, with unbound motifRMSD $\leq 1.0$ Å and bound motifRMSD $> 1.0$ Å.

- In both cases, we additionally required (i) motifRMSD standard deviation $\leq 0.5$ Å in each state to ensure the motif scaffolding/disruption was confidently predicted, and (ii) an absolute difference in mean motifRMSD between states $\geq 0.5$ Å to confirm a significant structural change.

Corresponding heatmaps of all 100 designs per motif and effector are visualized in Figure 6. Full scatterplots highlighting the bound vs. unbound average and standard deviation of motifRMSD are shown in Figures 7 and 8, with successful designs (according to the criteria above) colored pink. Example structures visualized in main text Figure 3.

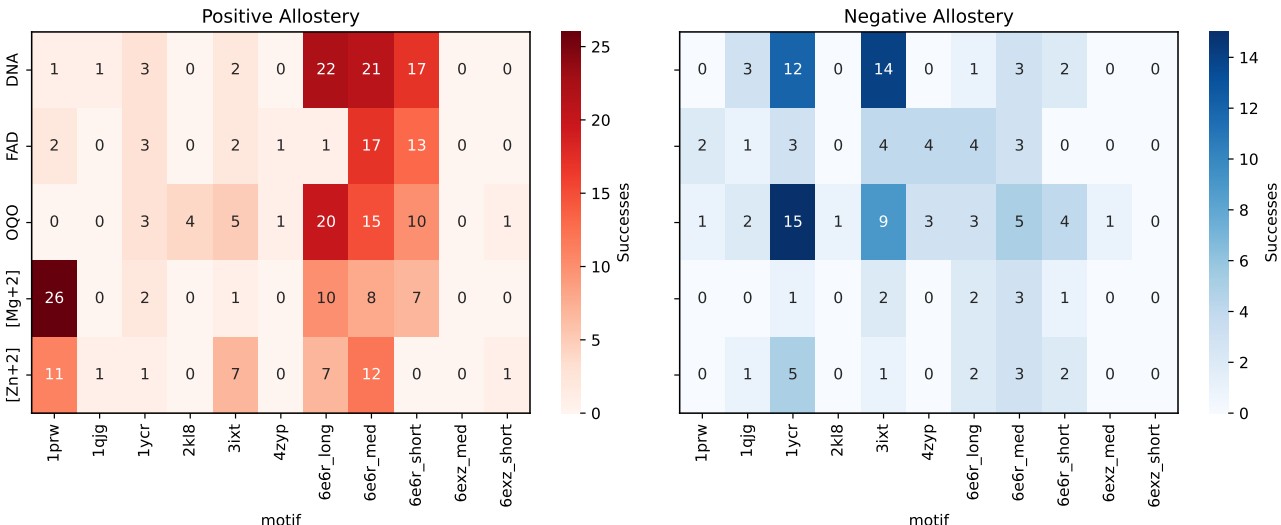

*Figure 6.* Success rates, out of 100 designs, for positive and negative allostery

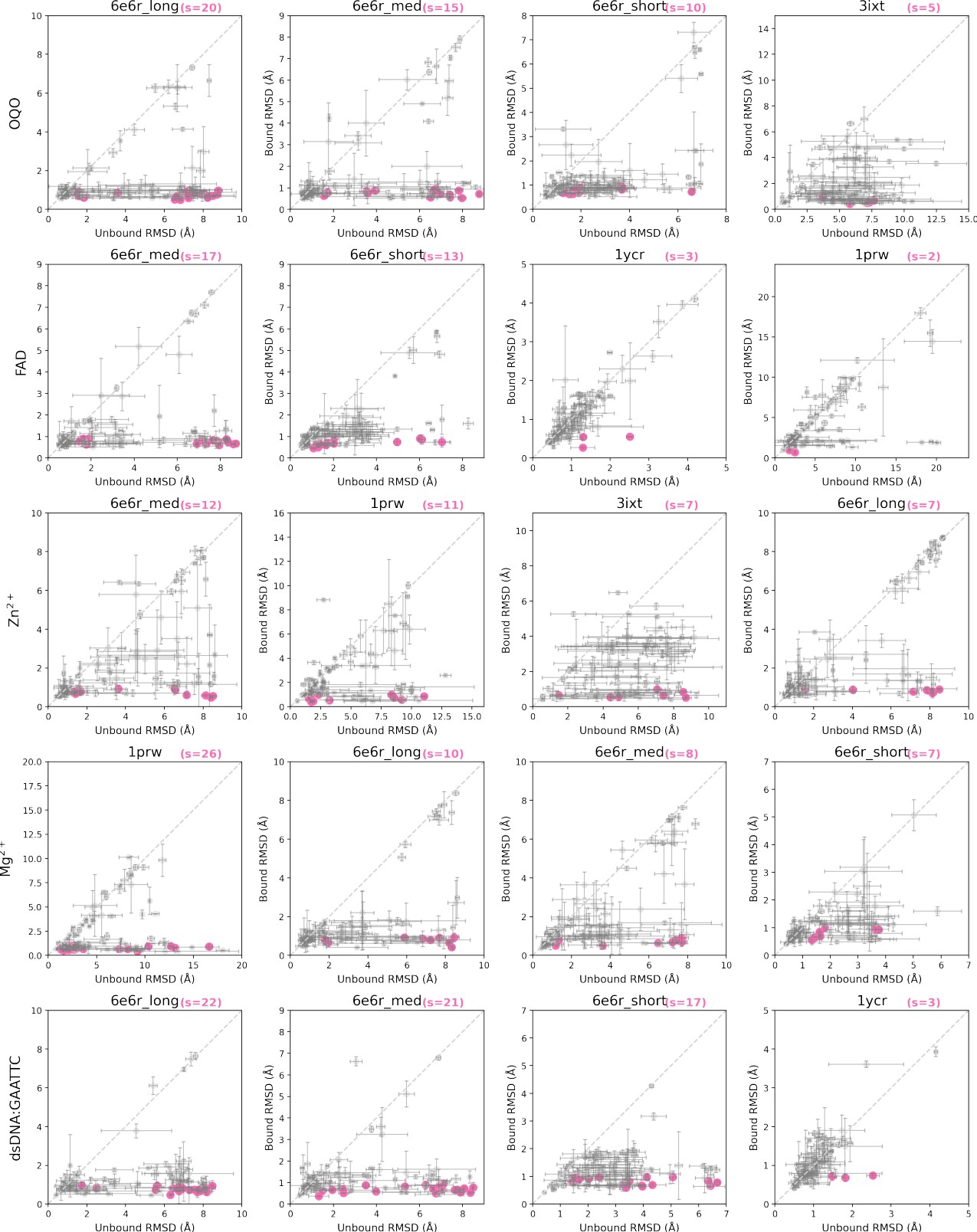

*Figure 7.* Positive allostery per effector/motif (top 4 successes) scatter plots visualizing bound (state B) vs unbound (state A) average motif RMSD. Error bars for each design quantify motif RMSD std. Successful designs colored in pink (corresponds to Fig 6 left).

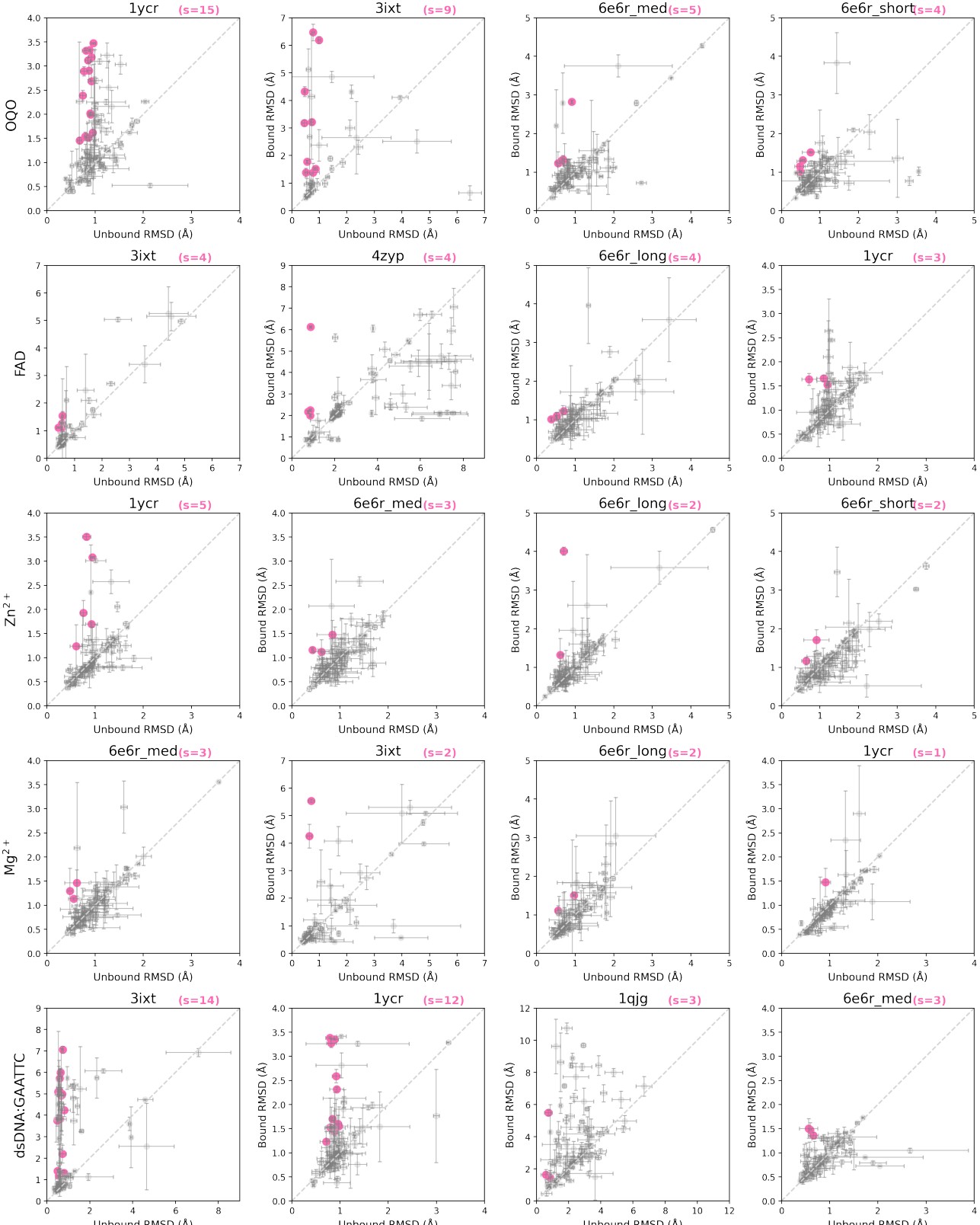

*Figure 8.* Negative allostery per effector/motif (top 4 success) scatter plots visualizing bound (state B) vs unbound (state A) average motif RMSD. Error bars for each design quantify motif RMSD std. Successful designs colored in pink (corresponds to Fig 6 right).

**Positive allostery of 3IXT with Mg$^{2+}$. 3.06 Å / 1.09 Å / 8.22 Å.**

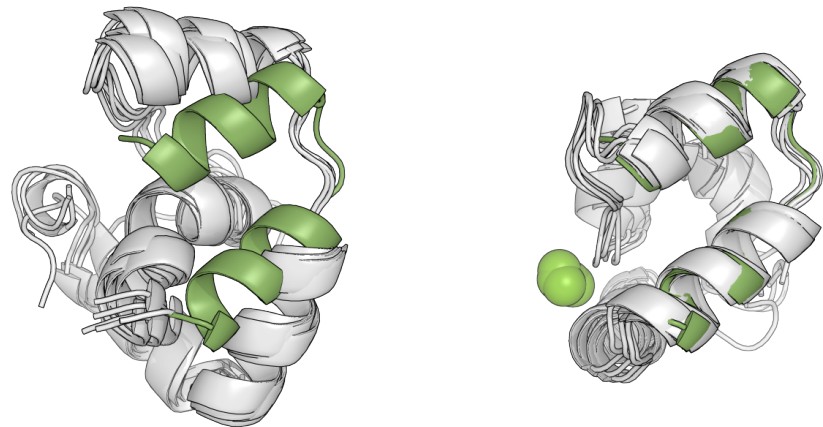

**Positive allostery of 2KL8 with OQO. 1.90 Å / 1.27 Å / 9.52 Å.**

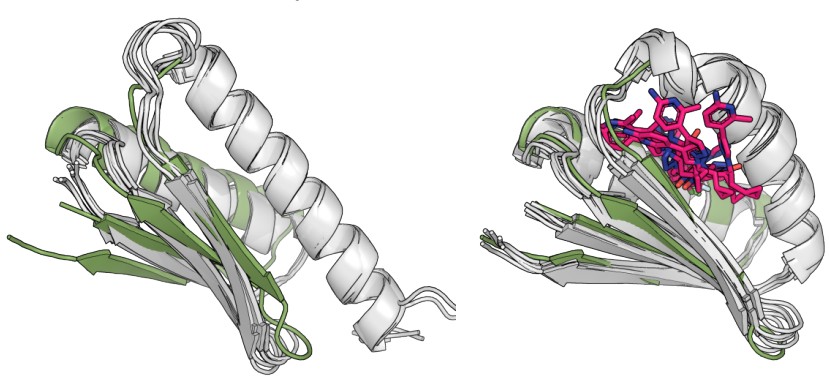

**Negative allostery of 1PRW with FAD. 0.60 Å / 0.68 Å / 1.78 Å.**

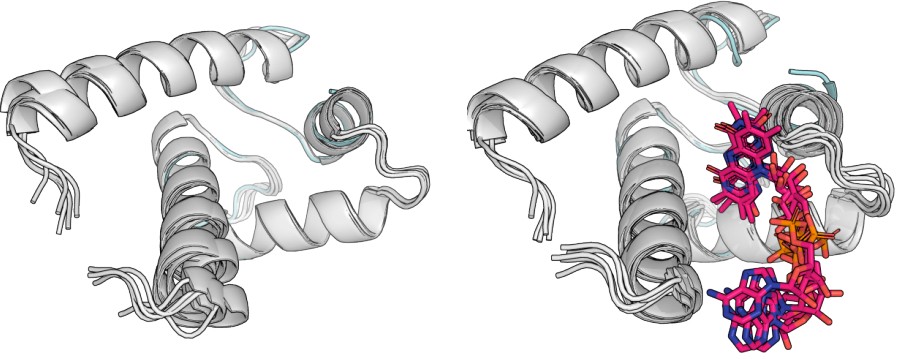

**Negative allostery of 1YCR with dsDNA. 4.40 Å / 1.96 Å / 17.04 Å.**

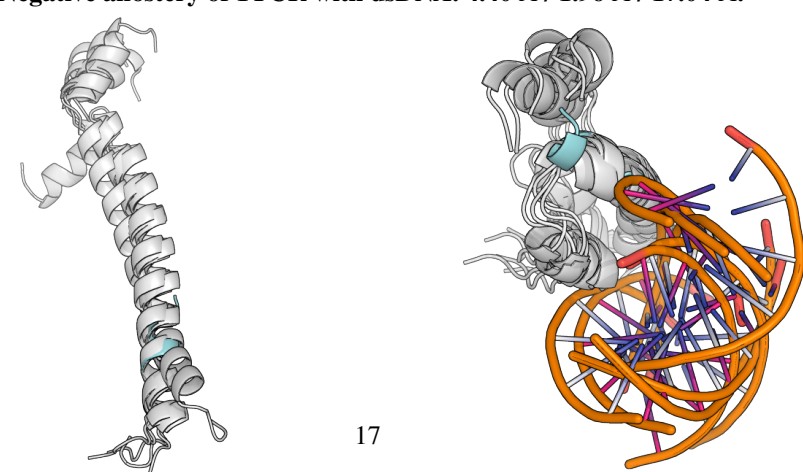

*Figure 9.* All predicted structures (5 each) for the positive and negative allostery designs highlighted in Figure 3. Intra-state RMSDs for the two states and cross-state Cα RMSDs are shown.

## B.2. Motif switching

We designed a scaffold containing two functional motifs, 3IXT and 1YCR, and generated 100 designs with the objective of reciprocal motif activity. Specifically, in the unbound state, the scaffold should present 3IXT in an active conformation while rendering 1YCR inactive, whereas in the OQO-bound state, the roles are reversed (1YCR active, 3IXT inactive). Thus, the following success criteria were used:

- In state A (unbound): 3IXT mean mRMSD $\leq 1.0$ Å and 1YCR mean mRMSD $> 1.0$ Å.

- In state B (OQO-bound): 1YCR mean mRMSD $\leq 1.0$ Å and 3IXT mean mRMSD $> 1.0$ Å.

- Additionally, mRMSD standard deviation $\leq 0.5$ Å for structure confidence.

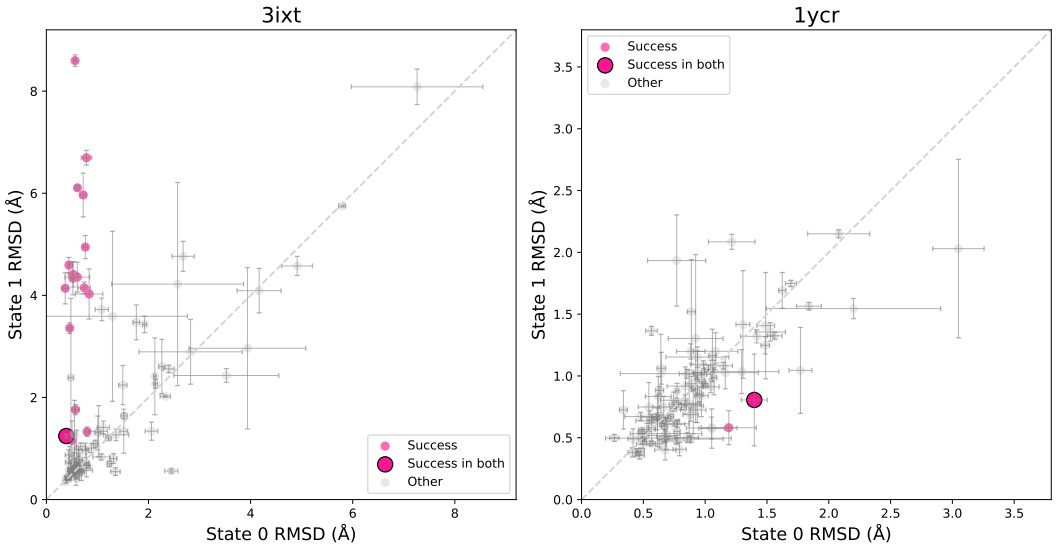

*Figure 10.* State B vs. State A mRMSD for each of the 100 3IXT/1YCR motif switching designs.

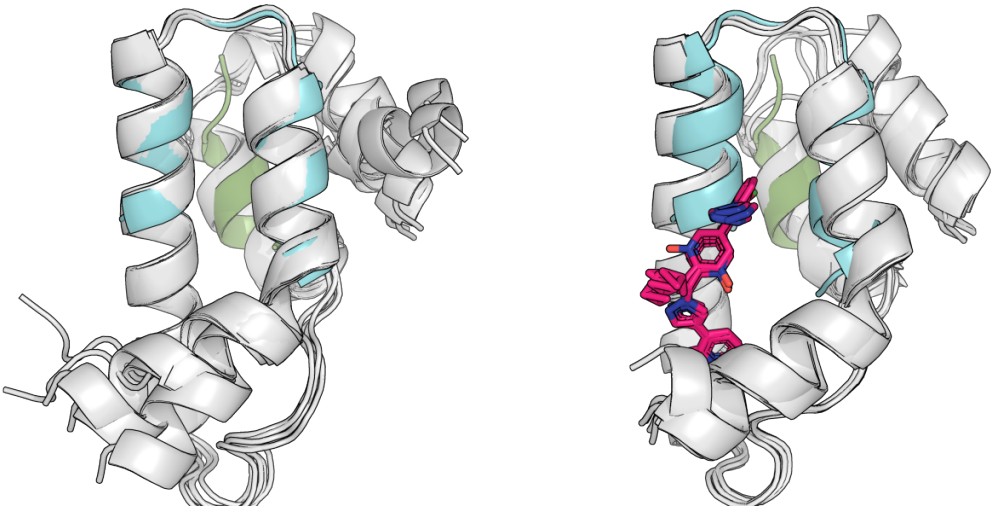

*Figure 11.* Predicted structures for the motif switching design. The **unbound** state A (*left*) has pLDDT=0.69, pTM=0.60, RMSD $1.93 \pm 0.74$ Å between samples, motif RMSD $0.39 \pm 0.02$ Å for 3IXT (blue), and motif RMSD $1.40 \pm 0.09$ Å for 1YCR (green). The **bound** state B (*right*) has pLDDT=0.79, pTM=0.80, RMSD $0.68 \pm 0.38$ Å between samples, motif RMSD $1.24 \pm 0.04$ Å for 3IXT (blue), and motif RMSD $0.68 \pm 0.38$ Å for 1YCR (green). The RMSD between states is $2.54 \pm 0.58$ Å. All RMSDs are computed for C$\alpha$ positions after alignment.

## B.3. Ligand modification

In the ligand modification task, we sought to design a protein with two states that differ only in the chemical form of their ligand. Specifically, in state A the protein binds heme, while in state B the heme ligand is oxygenated. The design goal was to capture distinct conformational changes upon ligand modification while maintaining stability within each state. A design was considered successful if it satisfied the following criteria:

- **Intra-state stability:** intra-state RMSD < 1.0 Å in the heme-bound state (A) or in the oxygenated-heme state (B).

- **Inter-state separation:** cross-state RMSD > 2.0 Å between states A and B (ensuring conformational distinction).

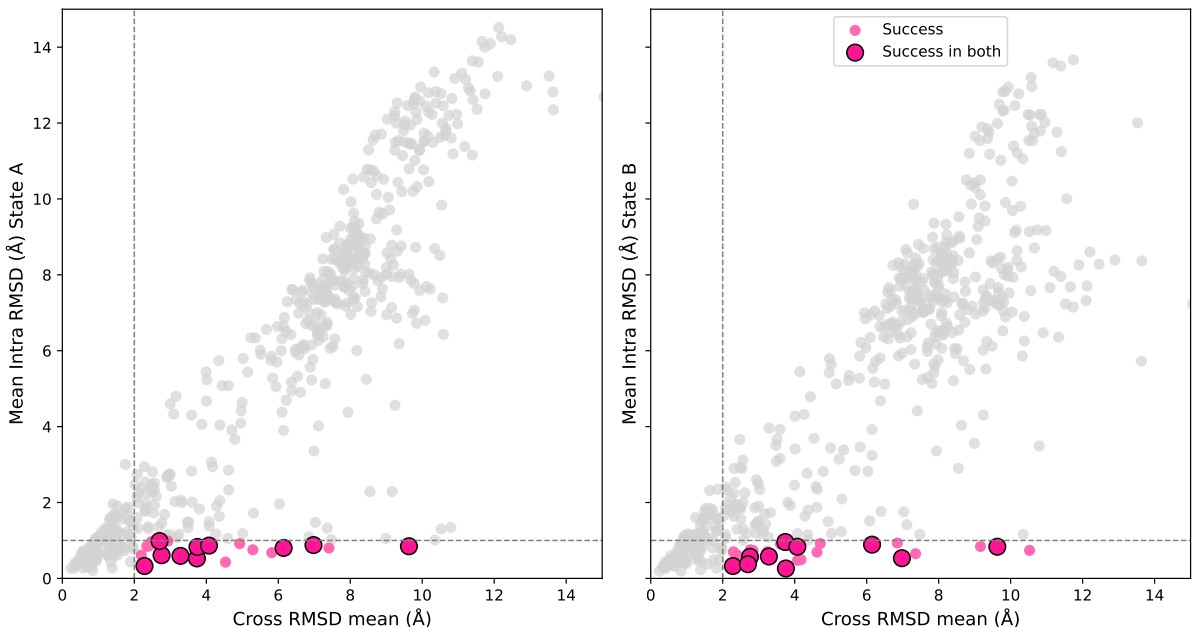

*Figure 12.* Success criteria for ligand modification designs. Each scatterplot shows cross-state RMSD (x-axis) versus intra-state RMSD (y-axis) for the heme-bound state A (*left*) and oxygenated-heme state B (*right*). Pink points indicate successful designs, with dark outlines denoting those that pass criteria in both states.

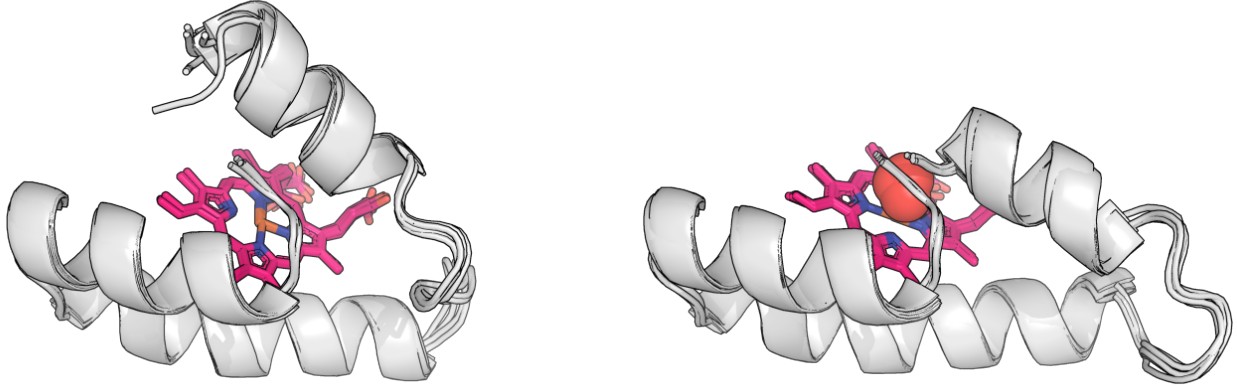

*Figure 13.* Predicted structures for the heme ligand modification design. The **unoxygenated** state A (*left*) has pLDDT=0.81, pTM=0.80, and RMSD $0.83 \pm 0.39$ Å between samples. The **oxygenated** state B (*right*) has pLDDT=0.91, pTM=0.94, and RMSD $0.26 \pm 0.05$ Å between samples. The RMSD between states is $3.76 \pm 0.08$ Å. All RMSDs are computed for C$\alpha$ positions after alignment.

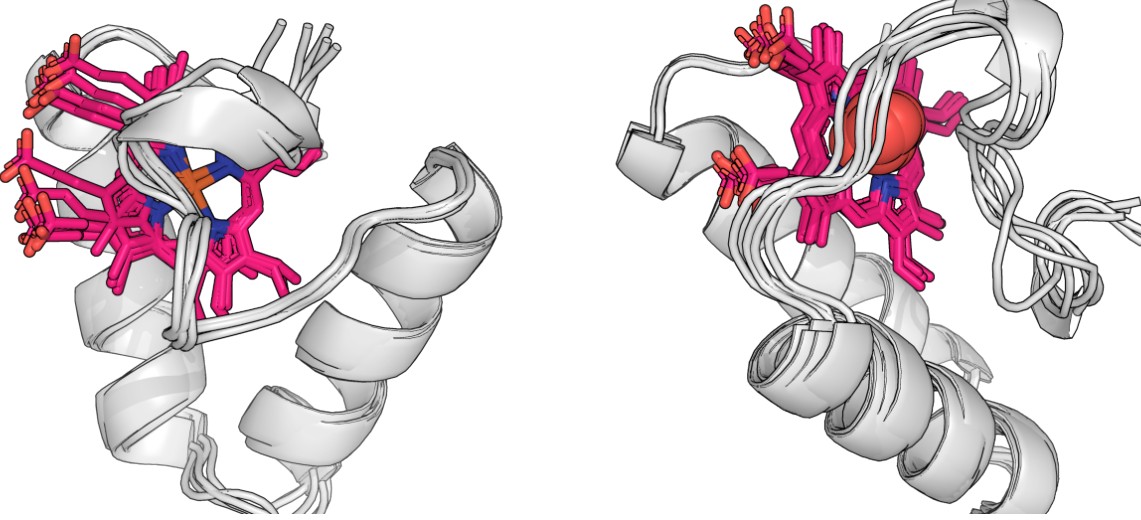

*Figure 14.* An alternative ligand modification design with a large but kinetically implausible conformational change. The **unoxygenated** state A (*left*) has pLDDT=0.74, pTM=0.68, and RMSD $0.85 \pm 0.18$ Å between samples. The **oxygenated** state B (*right*) has pLDDT=0.77, pTM=0.82, and RMSD $0.83 \pm 0.31$ Å between samples. The RMSD between states is $9.63 \pm 0.18$ Å.

## B.4. Induced binding

For the induced binding task, we wanted to design a protein that binds a Top7 fragment only in the presence of calcium, while remaining unbound in the calcium-free state. A design was considered successful if it satisfied the following conditions:

- **Binding specificity:** ipTM < 0.6 in the unbound state (no binding) and ipTM > 0.6 in the calcium-bound state (binding).

- **Conformational change:** inter-state RMSD > 3.0 Å between bound and unbound states.

- **Structural confidence:** intra-state RMSD < 2.0 Å within each state ensemble.

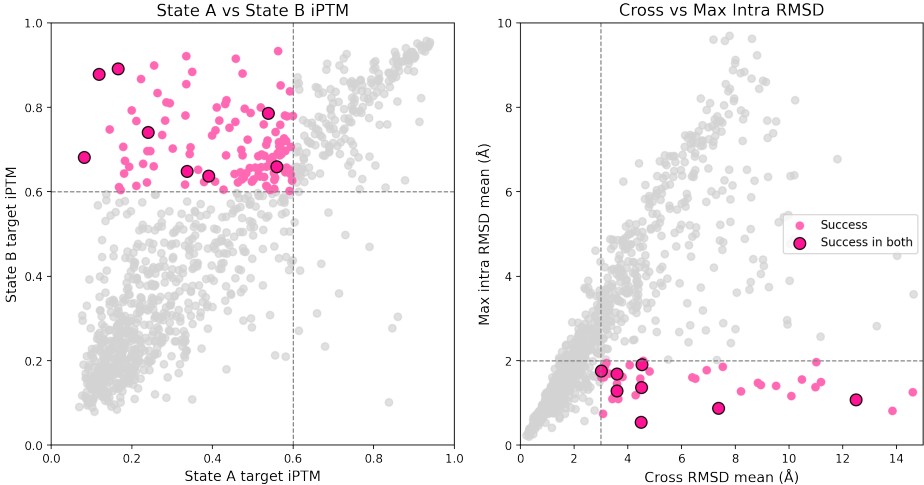

*Figure 15.* Success criteria for induced binding designs. (**Left**) ipTM in the unbound (state A) versus calcium-bound (state B) (**Right**) Cross-state versus intra-state RMSD. Pink points indicate designs meeting success thresholds, with dark outlines denoting designs that pass both sets of criteria.

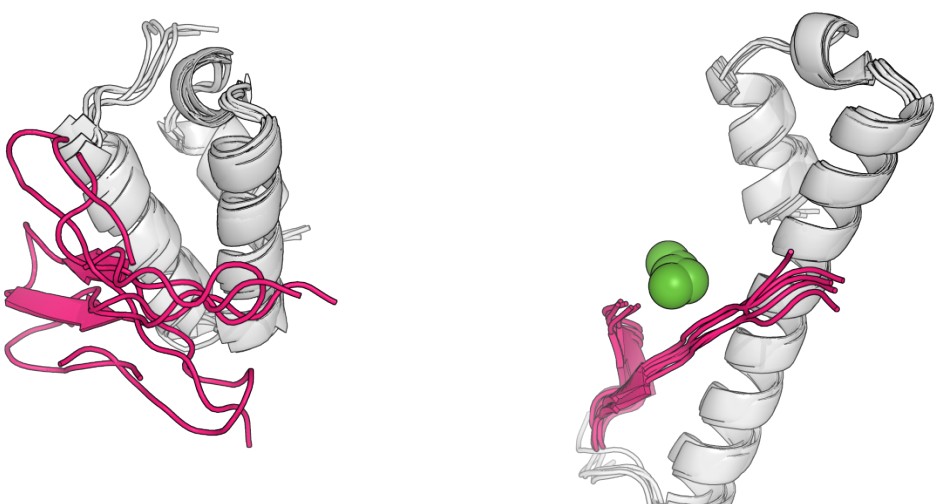

*Figure 16.* Predicted structures for the induced binding design. The **unbound** state A (*left*) has pLDDT=0.57, pTM=0.49, ipTM 0.12, and RMSD 1.08 ± 0.22 Å between samples. The **bound** state B (*right*) has pLDDT=0.85, pTM=0.80, ipTM 0.88 with respect to the target peptide, ipTM 0.62 with respect to the calcium effector, and RMSD 0.79 ± 0.14 Å between samples. The RMSD between states is 12.50 ± 0.17 Å. All RMSDs are computed for Cα positions of the design after alignment.

## B.5. Ligand discrimination

In the ligand discrimination task, we sought to design a protein with three distinct conformational states: (A) the **unbound** state, (B) the **OQO-bound** state, and (C) the **calcium-bound** state. Each state should be structurally stable on its own, while differing substantially from the other states to reflect distinct ligand-specific conformations. The success criteria are as follows,

- **Intra-state stability:** intra-state RMSD < 1.0 Å for all three states (ensuring consistent conformations within each ensemble).

- **Inter-state separation:** pairwise cross-state RMSD > 1.0 Å for all three pairs of states (ensuring distinct conformations between states).

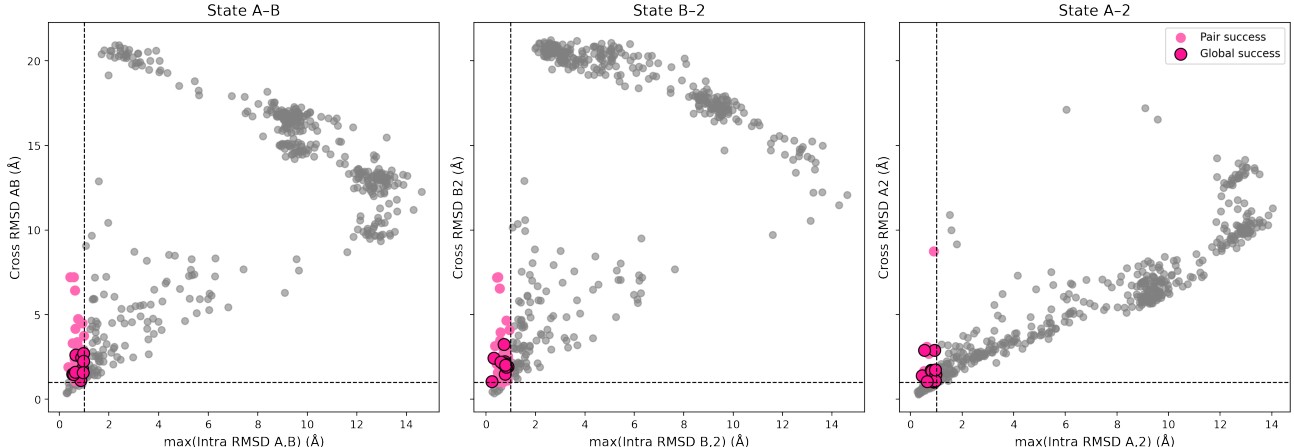

*Figure 17.* Success criteria for ligand discrimination designs. Each scatterplot shows the cross-state RMSD between a pair of states versus the maximum intra-state RMSD (where higher values indicate less consistency). Pink points indicate pairwise or global successes (outlined).

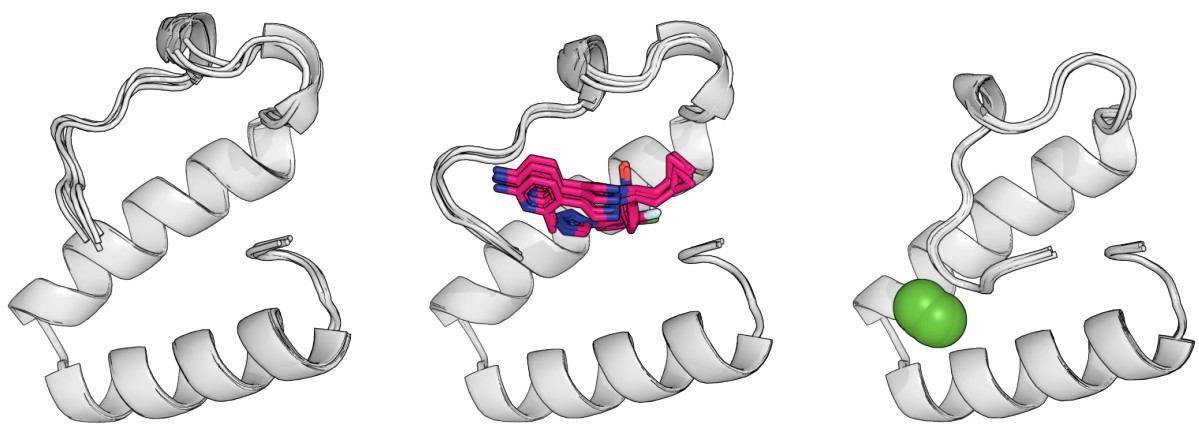

*Figure 18.* Predicted structures for the ligand discrimination design. The **unbound** state A (*left*) has pLDDT=0.75, pTM=0.56, and RMSD 0.53 ± 0.13 Å between samples. The **OQO-bound** state B (*center*) has pLDDT=0.85, pTM=0.90, and RMSD 0.31 ± 0.04 Å between samples. The **calcium-bound** state C (*right*) has pLDDT=0.86, pTM=0.89, and RMSD 0.21 ± 0.03 Å between samples. The RMSD between states is A/B: 1.48 ± 0.13 Å; A/C: 2.90 ± 0.10 Å; B/C: 2.42 ± 0.09 Å. All RMSDs are computed for Cα positions of the design after alignment.

## B.6. Controls & Ablations

In this section, we provide additional evidence comparing SwitchCraft to plausible controls and ablations.

To show that existing logit-tying approaches for state-switching design are inadequate, in Figure 19 we devise an off-the-shelf strategy for designing positive and negative allostery with RFD3 (Butcher et al., 2025) and TiedMPNN. This approach is generally unsuccessful compared to SwitchCraft (Table 1).

To validate that successful designs are resultant of the design objective (rather than a random baseline success rate), in Table 3 we tabulate the observed rates of *positive* allostery when designing for *negative* allostery, and vice versa. We observe that the choice of objective substantially increases the prevalence of the target behavior. We also validate that successful designs are less common when removing loss terms that promote state-switching behavior, i.e., the AntiMotifLoss (Table 2).

Finally, in Figure 20 we confirm that Boltz-1 rarely predicts multistate behavior among natural proteins, such that predicted successful designs are unlikely to be spurious hallucinations.

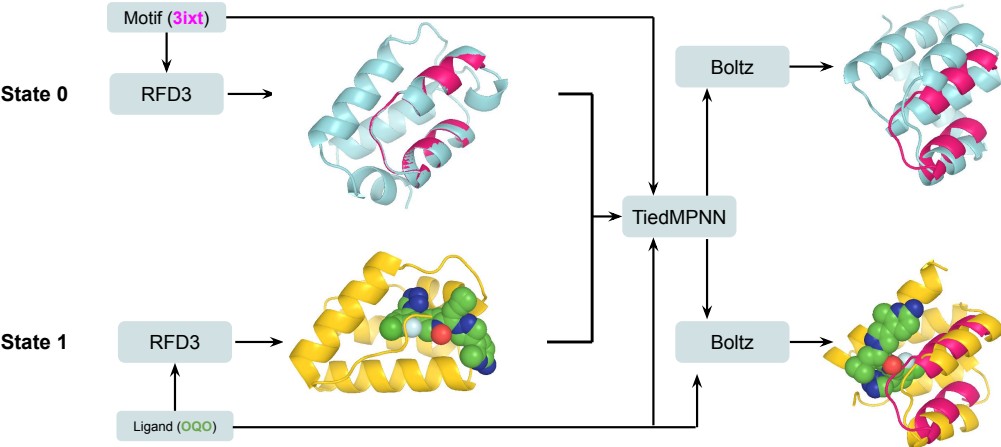

*Figure 19.* **RFD3+TiedMPNN fails to provide a sensible approach to multistate design** due to the incompatibility of input structures. In both cases, the resulting structures fail to scaffold the motif in either state due to the pollution of inverse folding logits from the highly incompatible alternate state.

*Table 1.* **SwitchCraft vs. RFD3+TiedMPNN on positive allostery**. Since no standard multistate baseline exists for this setting, we devised our own way to repurpose RFD3 to obtain sensible multistate results (see Figure 19).

| Effector | Motif | SwitchCraft | RFD3+TiedMPNN |
|---|---|---|---|
| FAD | 1prw | 2 | 2 |
| FAD | 1ycr | 3 | 3 |
| FAD | 3ixt | **2** | 1 |
| FAD | 6e6r_long | 1 | 1 |
| FAD | 6e6r_med | **17** | 0 |
| FAD | 6e6r_short | **13** | 4 |
| $Mg^{2+}$ | 1prw | **26** | 0 |
| $Mg^{2+}$ | 1ycr | 2 | 2 |
| $Mg^{2+}$ | 3ixt | **1** | 0 |
| $Mg^{2+}$ | 6e6r_long | **10** | 0 |
| $Mg^{2+}$ | 6e6r_med | **8** | 0 |
| $Mg^{2+}$ | 6e6r_short | **7** | 0 |
| OQO | 1prw | 0 | **4** |
| OQO | 1ycr | **3** | 2 |
| OQO | 3ixt | **5** | 2 |
| OQO | 6e6r_long | **20** | 1 |
| OQO | 6e6r_med | **15** | 2 |
| OQO | 6e6r_short | **10** | 2 |
| $Zn^{2+}$ | 1prw | **11** | 0 |
| $Zn^{2+}$ | 1ycr | 1 | 1 |
| $Zn^{2+}$ | 3ixt | **7** | 0 |
| $Zn^{2+}$ | 6e6r_long | **7** | 1 |
| $Zn^{2+}$ | 6e6r_med | **12** | 2 |
| $Zn^{2+}$ | 6e6r_short | 0 | **1** |

*Table 2.* **Results comparing SwitchCraft with and without AntiMotifLoss for positive allostery.** Results indicate AntiMotifLoss is indeed necessary to supervise motif disruption.

| Ligand | Motif | SwitchCraft | SwitchCraft No AntiMotifLoss |
|---|---|---|---|
| OQO | 3ixt | **5** | 1 |
| OQO | 6e6r_long | **20** | 6 |
| OQO | 6e6r_med | **15** | 5 |
| OQO | 6e6r_short | **10** | 2 |
| $Zn^{2+}$ | 3ixt | **7** | 3 |
| $Zn^{2+}$ | 6e6r_long | **7** | 1 |
| $Zn^{2+}$ | 6e6r_med | **12** | 1 |
| $Zn^{2+}$ | 6e6r_short | 0 | 0 |

*Table 3.* **Cross screening of allostery designs**. Designs for positive or negative allostery (same number for each) are screened for the target behavior and the opposite behavior. If the design choices were ineffective, the rates of each type of behavior would be the same across design settings.

| Design setting | Observed behavior | |
| --- | --- | --- |
| | Positive allostery | Negative allostery |
| Positive allostery | **259** | 22 |
| Negative allostery | 86 | **123** |
| Fold change | 3.01x | 5.59x |
| *p*-value | $2.9 \times 10^{-21}$ | $3.2 \times 10^{-18}$ |

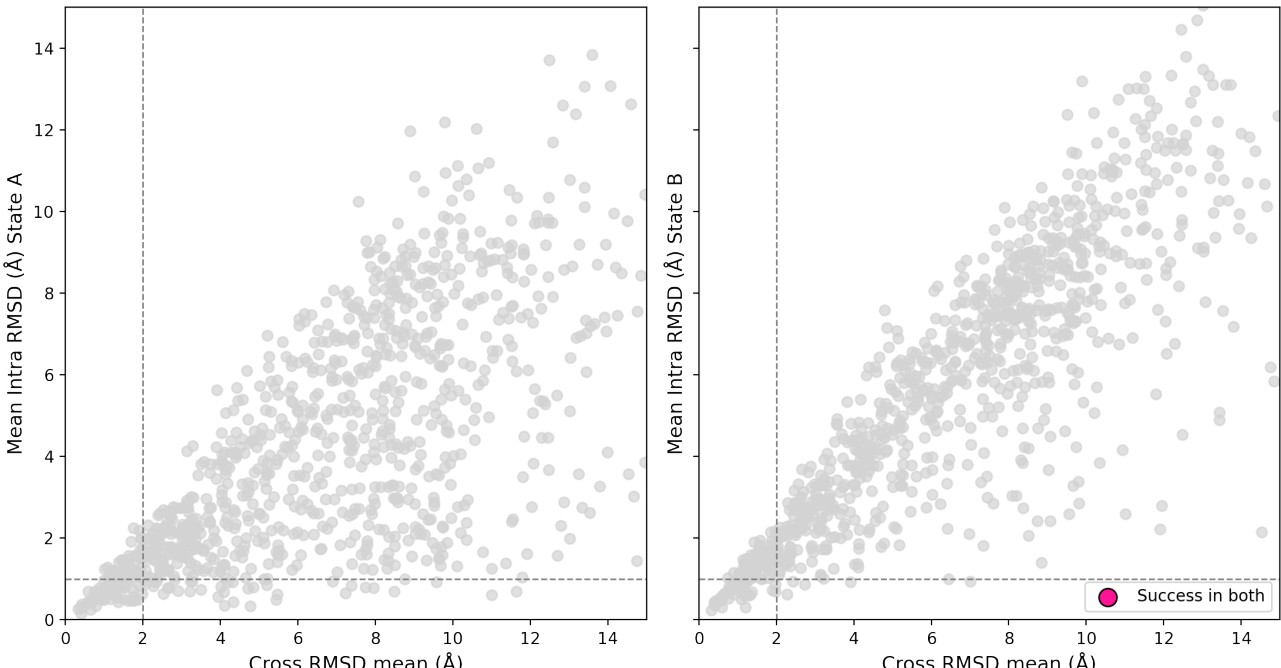

*Figure 20.* **1000 random UniRef50 miniproteins cofolded with random CCD ligands.** No proteins pass thresholds for confidently predicted conformational change (cross RMSD > 2 Å, intra RMSD < 1 Å in both states).

## B.7. Preliminary wet lab validation

*Table 4.* **Initial wet lab validation of SwitchCraft designed zinc induced PD-L1 binders.** Binding affinities were measured by BLI in triplicate across increasing $Zn^{2+}$ concentrations to evaluate zinc-dependent modulation. Both designs exhibit conditional activity, with no detectable PD-L1 binding in the absence of zinc and measurable binding upon zinc addition.

| | $Zn^{2+}$ concentration | | | | | |
| Design | 0 $\mu$M | 15 $\mu$M | 20 $\mu$M | 50 $\mu$M | 100 $\mu$M | 200 $\mu$M |
|---|---|---|---|---|---|---|
| 2874 | No binding | Detected, n.r. | $K_D = 1.9\ \mu$M | $K_D = 2.9\ \mu$M | $K_D = 2.6\ \mu$M | $K_D = 3.0\ \mu$M |
| 278 | No binding | Detected, n.r. | Detected, n.r. | $K_D = 746$ nM | $K_D = 4.8\ \mu$M | $K_D = 1.0\ \mu$M |

n.r., binding detected but $K_D$ not resolved.

