# OpenReview forum: "SwitchCraft: A Programmatic Framework for Designing State-Switching Proteins"
_ICML.cc/2026/Conference — ICML 2026 regular_

### Official Review · Reviewer_ELDd · 2026-03-12

**Soundness:** 2
**Presentation:** 2
**Significance:** 3
**Originality:** 3
**Overall Recommendation:** 3
**Confidence:** 5

**Summary:**

This paper proposes SWITCHCRAFT, a programmatic framework for multistate protein design. The core idea is to specify multiple functional states as compositional losses, then optimize a single sequence jointly across these states by backpropagating through a structure prediction model. Concretely, the framework defines losses such as motif scaffolding, ligand binding, anti-binding, conformational change, and contact preservation, all parameterized primarily by Boltz-1. The sequence is represented by logits and optimized using a multi-stage schedule adapted from BoltzDesign1. Experiments cover six multistate design primitives, including positive/negative allostery, motif switching, ligand modification, induced binding, ligand discrimination, and a biosensor-design case study. The paper argues that this provides a general computational framework for multistate protein design.

**Compliance With Llm Reviewing Policy:**

Affirmed.

**Final Justification:**

I acknowledge that I have read the author rebuttal. I understand that Area Chairs may flag insufficient reviews during the Reviewer-AC Discussion period and shortly thereafter to address irresponsible, insufficient, or otherwise problematic reviewer conduct. Area Chairs will also be able to flag up during Metareview grossly irresponsible reviewers (including but not limited to neglect of review duties). I understand that my review and my conduct are subject to the ICML Peer Review Ethics (https://icml.cc/Conferences/2026/PeerReviewEthics), and that grossly irresponsible behavior may result in the desk rejection of my co-authored papers. I increased my score to 3.

**Key Questions For Authors:**

1. The method and the evaluation both rely almost entirely on Boltz-1 outputs. In Section 3.1, even the conformational-change objective is defined by maximizing JSD between Boltz-1 distograms across states. How do the authors disentangle true sequence-encoded multistability from Boltz-1-specific context sensitivity or sampling noise?

2. Could the authors provide independent validation of the final designed sequences using third-party structure predictors or other external tools, especially for Sections 4.2 to 4.5? At present, designs are optimized and then re-evaluated within the same Boltz-1 loop.

3. Why was there no recovery-style control experiment on known multistate proteins? For example, if the two states of a known allosteric protein are provided, can SWITCHCRAFT recover native-like sequences or at least native-like multistate logic? Conversely, for a protein known not to be allosteric, does the framework still tend to manufacture state differences?

4. Section 3.2 uses a heavily scheduled optimization procedure with four stages and several interacting hyperparameters. What will happen if you keep the same hyperparameters for all designs?

5. line 428 has two 'and'

6. The basic assumption of this work is that Boltz1 can do multi-state sampling and the predictions are accurate, but it seems that there is no clear proof.

7. It seems that the codes or dataset are not open-sourced.

**Limitations:**

I do not think the limitations are discussed sufficiently.

1. The framework is strongly dependent on Boltz1 both as the design objective engine and as the evaluator of success. This creates a substantial closed-loop bias and weakens confidence that the results would generalize under independent validation.

2. The conformational-change objective is based on predicted distogram divergence rather than on a physical model of multistate accessibility, stability, or kinetics.

3. The optimization pipeline is heavily heuristic. The method is programmatic at the level of loss composition, but still fairly hand-engineered at the level of optimization schedule and task-specific design heuristics.

4. The current experiments don't have the controls I would like to see, such as known multistate recovery, non-allosteric negative controls, and independent external re-evaluation of final sequences.

**Strengths And Weaknesses:**

Soundness:

Strengths: the problem is important, the framework concept is attractive, and the paper contains substantial engineering work. I also think the formulation is internally coherent: the method clearly defines states, losses, and a sequence-optimization loop, and the experiments show that the proposed objectives can indeed find sequences that satisfy the paper’s in silico criteria across several multistate settings.

Weakness: the entire framework is too tightly coupled to Boltz-1. In Section 3.1, the losses are defined almost entirely on top of Boltz-1 outputs, including motif loss, binding loss, and especially conformational change loss, which maximizes JSD differences between Boltz-1 distograms across states. This means the method relies on Boltz-1 not only for optimization, but also for interpreting whether a design has the intended multistate behavior. If Boltz-1 is not reliable for ligand-dependent conformations or multistate sampling, then the losses themselves become unreliable proxies.

I am also not fully convinced by the conformational-change objective. The paper defines it by maximizing, for each residue, the most susceptible contact’s distogram JSD across two states. I understand why this is convenient and differentiable, but it is not obvious to me why maximizing the most susceptible predicted contact per residue should correspond to a meaningful or physically realizable conformational change (what is 'susceptible'?). More importantly, it encourages predictive distribution differences, not necessarily kinetically accessible or thermodynamically plausible state transitions. The authors themselves partly acknowledge this in Section 4.3, where some designs appear to require implausible unbinding and rebinding, and they suggest future kinetics-based constraints.

Section 3.2 is another weakness for me. The optimization is not simple, but a heavily scheduled heuristic procedure with multiple hyperparameters. This may work in practice, but it makes the framework feel fairly hand-engineered, and it is unclear how much of the performance comes from the design objectives versus the optimization heuristics.

The experiments also remain too closed-loop. For all settings the final sequence is evaluated by predicting five structures with Boltz-1 in each state and measuring quantities.  If I'm correct, I didn't see any independent validation using a third-party predictor or physics-based method. As a result, the paper currently demonstrates that SWITCHCRAFT can find sequences that Boltz-1 believes are multistate, but not yet that the framework robustly captures true multistate behavior or really work on other situations.

Presentation:

Strengths: the paper is clearly written overall, and the high-level story is easy to follow. I think the authors do a good job motivating why multistate design is important and why existing single-state tools are insufficient. Figure 1 is clear, and Figure 2 usefully summarizes the design primitives.

Weakness: the paper feels more complicated than necessary, partly because the case-study figures are emphasized early, while the overall quantitative summaries come later. In particular, Figure 3 and Figure 4 appear before the larger statistical summaries they depend on, which makes the narrative feel case-driven. I also think the success criteria should be stated more plainly in the main text, since many claims depend on heuristic thresholds.

Significance:

Strengths: I think the paper addresses an important problem. Multistate protein design is clearly relevant and underdeveloped compared with single-state scaffold or binder design. A framework that lets users specify multiple states and optimize a single sequence across them could be very useful if it becomes more reliable and easier to use.

Weakness: I currently view the significance as more potential than demonstrated. At present, the paper shows impressive in silico examples inside a single model loop, but not yet enough evidence that the framework is reliable, general, or externally validated.

Originality:

Strengths: the paper is reasonably original at the framing level. The idea of expressing multistate function as compositional losses over multiple folding contexts, then optimizing one sequence jointly across states, is interesting and has good potential.

Weakness: the method novelty is somewhat limited by the heavy reuse of Boltz-1 and BoltzDesign1. The losses are mostly Boltz-1-based proxies, and the optimizer is adapted directly from BoltzDesign1 with a fairly elaborate hand-tuned schedule. So I see this as an original and promising framework idea, but not yet a mature or strongly validated method.

---

> ### Author Rebuttal · Authors · 2026-03-29
>
> Thank you for your detailed review. We are glad to hear that you deem our problem important, the framework attractive and original, and the engineering substantial. We also appreciate your concerns and look forward to discussing them together. **All fig/tables referenced are available at https://tinyurl.com/mrxy4utf**
>
> ---
>
> ### **Further validation**
>
> We provide new evidence that our predicted conf changes are independently verifiable, are not spurious, and are the direct consequence of our design protocol.
>
> **Independent oracle**
>
> We use OpenFold3, an independently trained cofolding model, to re-predict structures in the neg allostery and induced binding design tasks (linked Tab3-4). OpenFold3 verifies the presence of succesful designs under identical thresholds, although the exact success rate varied (higher for induced binding, lower for neg allostery).
>
> **Wet-lab validation**
>
> Additionally, although not within the scope of this submission, we report ongoing experimental validation for your consideration. We have designed and experimentally verified (1) a _de novo_ minibinder with function activated by an external analyte (linked Fig1) , (2) allosteric deactivation of a wild-type enzyme (3) _de novo_ biosensor candidates that modulate fluorescence differently than WT GFP in the presence of the target analyte. Our experimental results will be reported in a separate manuscript; however, we offer these results to provide further support of the merits of our computational method.
>
>
> **Prediction of native conf changes**
>
> Cofolding models can indeed predict multistate behavior; (doi.org/10.64898/2026.01.04.697564) showed that AF3 can model cryptic pockets opening in response to ligand binding. With Boltz, we were able to recapitulate several similar examples (linked Fig2). These give us confidence that cofolding models can model how protein conformations are dependent on molecular contexts.
>
> Perfect predictive ability is not required for a method to be used in screening, as long we **filter for high-confidence predictions** (which we do). In this same fashion, cofolding models are widely used for evaluating de novo binders, even though PPI prediction is unsolved.
>
> **Negative controls**
>
> - To verify that predicted multistate behavior is not spurious, we design single state motif scaffolds and fold them w/wo ligands to check for motif disruption (linked Tab6). Few resulting designs pass our thresholds for negative allostery, indicating that designs do not generally exhibit state-switching.
>
> - We verify that pos allostery is far more frequent when optimizing for pos than neg and vice versa (linked Tab5). If the observed multistate behavior were spurious, these rates would be similar across design conditions.
>
> - We re-run pos allostery designs w/o AntiMotif loss (which promotes conf change between states); far fewer designs succeed (linked Tab2).
>
> - We run Boltz on wild-type neg controls by cofolding 1000 random miniproteins (length 50 crops) from UniRef50 with randomly paired ligands from the CCD; none of these (linked Fig4) passed our thresholds for a confidently predicted conf change (<1 A intraRMSD, >2 A crossRMSD).
>
> These results strongly suggest that Boltz is not prone to manufacturing state-switching behavior when it is not present or explicitly designed for.
>
> ---
>
> ### **Recovery of native multistate behavior**
>
> We emphasize that our design tasks were chosen to reflect common multistate behavior in natural proteins.
> - induced binding (Ca2+ induced binding of calmodulin to CAMK)
> - ligand modification (hemoglobin cooperativity)
> - ligand discrimination (GPCR biased signaling)
> - biosensor design (quenching of chromophore PDB 7s7u/7s7v)
>
> ---
>
> ### **Conf change loss**
>
> We clarify that we only use the conf change loss in the final two design settings (ligand discrimination and modification); in all other scenarios, the difference between states is encoded by the motif and anti-motif losses. Hence, we do not seek to emphasize the exact form of the loss; rather the concept of a toolbox of loss functions to unlock new design capabilities.
>
> In a well-predicted structure, the distogram is sharply peaked for most residue pairs; hence, the JSD loss pushes the peaks to different values between states. The "susceptible" residue pairs are simply those which appear likely to change in distance between the two states.
>
> ---
>
> ### **Optimization procedure**
>
> The hyperparameters of our optimization procedure are **identical across all design settings**, without any tuning, and taken from prior work (BindCraft; BoltzDesign). Hence, we disagree that our optimization schedule is "hand-engineered" and involves "task-specific heuristics."
>
> ---
>
> ### **Summary**
>
> We have strengthened our results by (1) supplying independent evaluations (2) verifying multistate modeling and (3) running numerous neg controls. We also hope to have clarified misunderstandings of our (1) conf change loss and (2) optimization procedure.

---

> > ### Author Rebuttal · Reviewer_ELDd · 2026-04-01
> >
> > I acknowledge that I have read the author rebuttal. I understand that Area Chairs may flag insufficient reviews during the Reviewer-AC Discussion period and shortly thereafter to address irresponsible, insufficient, or otherwise problematic reviewer conduct. Area Chairs will also be able to flag up during Metareview grossly irresponsible reviewers (including but not limited to neglect of review duties). I understand that my review and my conduct are subject to the ICML Peer Review Ethics (https://icml.cc/Conferences/2026/PeerReviewEthics), and that grossly irresponsible behavior may result in the desk rejection of my co-authored papers.
> > I increased my score to 3.

---

> > > ### Author Response · Authors · 2026-04-04
> > >
> > > We are thrilled to hear that your concerns have been fully resolved! We are thankful for your comments, which have improved the presentation of our paper. If you would support acceptance, would you kindly consider increasing your score to "Weak Accept" or "Accept"? Thank you!

---

### Official Review · Reviewer_ci2s · 2026-03-12

**Soundness:** 2
**Presentation:** 3
**Significance:** 2
**Originality:** 1
**Overall Recommendation:** 3
**Confidence:** 4

**Summary:**

The paper describes an extension of BoltzDesign-1 to ligand-inducible state-switching protein design via appropriate choices of differentiable loss functions to be used for gradient descent on Boltz-1 sequence inputs. The main contributions of the paper are the specific set of loss terms used for multi-state design, including a possibly novel conformational change loss term, as well as the empirical investigation of the performance of the resulting design strategy on a series of constructed in silico multi-state design problems.

**Compliance With Llm Reviewing Policy:**

Affirmed.

**Key Questions For Authors:**

1) Empirically, the paper suffers from (i) a lack of contextualisation of the reported in silico metrics (are the success rates good or bad?) and (ii) a lack of insight into the strengths and weaknesses of the proposed method. For example, the main identifiable point of difference to BoltzDesign-1 is the conformational change loss function, but no empirical exploration of the form of this function is provided. Would removing the conformational change loss function work just as well? What are the critical parameters to tune to make multi-state design work? How do they depend on the type of problem? Are there any interesting differences from the parameters required to make single-state design work?

2) Presumably, there is a non-zero base rate for conformational changes when cofolding the same protein with and without the ligand. The paper would benefit from a comparison of the success rates on the conformational change metric with and without the conformational change loss

3) The paper claims that no existing methods are applicable to the type of multi-state design they consider, however I would be inclined to question this. Similar to how they have repurposed BoltzDesign to multi-state design, it should be possible to repurpose structure generation + inverse folding pipelines (like RFDiffusion) to design backbones given structural contexts, then design sequences using a logit-tied ligandMPNN. A paper comparing the suitability of generative and hallucination style pipelines to multi-state design might offer some insights; without this kind of comparison it is hard to know how to interpret the scientific value of the in silico design results.

4) Another meaningful contribution that could be made in this area would be to provide a benchmark of multi-state design problems. The authors make some progress towards this by constructing a relevant set of problems, however significantly more work in defining a broad,
standardised set of relevant problems and offering some clues as to expected or baseline performance on them would be required for this kind of contribution.

**Limitations:**

yes

**Strengths And Weaknesses:**

Strengths:
1) The paper is clearly written and provides succinct and accurate summaries of relevant literature.

2) The figures demonstrating the design problems are very nice. The problem setting is an interesting and important one, and the choices of specific multi-state design tasks seem well-considered.

3) In the supplementary additional results section, we can see that switching the objective between positive and negative allostery shifts the distribution (this is the one case where we have strong empirical evidence that allows us to see how much the loss is changing the generated structures) it would still benefit from an ablation where we see the success rates with no loss.

4) conformational change loss is novel

5) Constructing loss terms from multiple states (with and without ligand) is novel



Weaknesses:

1) In current form, the paper unfortunately makes an insufficient methodological or empirical contribution. Methodologically it is a straightforward extension of existing tools (perhaps even a direct application of existing tools to a slightly non-standard problem). It is interesting but unsurprising to see methods that work for single-state design being applied to multi-state design, and there is no further insight emerging from the results about specificities of the multi-state setting.

2) lack of baselines to contextualise the performance

3) no empirical exploration of the form of the conformational change loss function

At present, the paper is more suitable as a workshop paper.

---

> ### Author Rebuttal · Authors · 2026-03-28
>
> Thank you for the review. We’re pleased that you view our problem setting as important and our execution well-considered. We are happy to clarify the contributions of our work and further contextualize our results. **All fig/tables referenced are available at https://tinyurl.com/mrxy4utf**
>
> ---
>
> ### **Key contributions**
>
> The challenge in converting functional protein design to a deep learning problem is the **programmatic description** of complex functions and a **design protocol** to obtain sequences to satisfy these programmatic descriptions.
>
> Our work has provided significant advancements in both. We are pleased that you find the framework clear in hindsight; however, from a broader perspective, our contribution is substantial and nonobvious.
>
> Methodologically, it is nontrivial to accomplish programmatic functional design with existing techniques. It **can not** be framed as
> * Conditional generative modeling (insufficient training data).
> * Backbone generation + inverse folding with tied logits. **Logit-tying is not a panacea**; we elaborate below.
>
> Our key insight is that programmatic multistate design could be best accomplished with multi-objective optimization, building on top of backprop-based binder design. This had not been recognized or implied in prior literature; hence, **in the most objective sense, our contribution is novel.**
>
> Empirically, our framework is the first to demonstrate how to design higher-order protein functions from scratch. While many have focused on eliciting functional generation via natural language or GO terms, our work describes complex functions via a more principled approach. This has enabled us to access design targets (allostery; heme oxygenation; biosensors) far beyond prior capabilities.
>
> > "interesting but unsurprising"
>
> We disagree that a contribution must be "surprising" to the reviewer to be worthy of publication. Many key advances in our field were unsurprising or even obvious to colleagues of the authors, but were surprising to broader audiences interested in the results but not familiar with the approach. _If you think such an audience exists for our work, we encourage a vote for acceptance._
>
> ---
>
> ### **Comparison with structure generation pipelines**
>
> Although the proposal to "design backbones given structural contexts, then design sequences using a logit-tied ligandMPNN" seems sensible, this hypothetical framework has several issues.
>
> First, the intended use case of tied MPNN is for designing symmetric multimers, where the input backbones are identical. Separately designed backbones are generally too dissimilar to be satisfied by a single sequence (linked Fig3). Coupling backbones to ensure their compatibility may be possible, but such a technique would merit its own paper.
>
> Second, it is unclear how to control the differing _behavior_ of backbones generated with different conditioning. For example, backbone generation cannot provide control over (1) anti-binding behavior, i.e. sampling p(struct | NOT ligand); (2) "off" motif states, i.e., sampling p(struct | NOT motif). SwitchCraft can handle these cases by inverting binding and motif lossess.
>
> To make these points clear, we run RFD3 on the pos allostery task for several ligand / motif pairs. We sample two backbones: one conditioned on the motif, one conditioned on the ligand, and run tied MPNN. Of 24000 designs, much fewer passed our success threshold compared to SwitchCraft (linked Tab1).
>
> ---
>
> ### **Conformational change loss**
>
> We believe there may be a misunderstanding of of the role of our conformational change loss. We only use it in the final two design settings; in all other scenarios, the difference between states is driven by **the application of different losses to different states**. Hence, we emphasize the judicious selection of loss functions to encode conceptual design targets rather the exact form and hyperparameters of any particular loss function, which should be optimized on a case-by-case basis.
>
> > the main identifiable point of difference to BoltzDesign-1 is the conformational change loss function
>
> The motif loss is also novel; BoltzDesign1 did not demonstrate any motif scaffolding.
>
> ---
>
> ### **Contextualization of performance**
>
> To establish the baseline rate of multistate behavior, we design single-state motif scaffolds and screen for disruption of the motif upon ligand binding (e.g. neg allostery). Across 2000 total designs, we find 21 passing our criterion, versus 197 for SwitchCraft's multistate protocol in the same setting (linked Tab6). This establishes that predicted multistate behavior is rare when not explicitly designed for.
>
> To study the contribution of the losses promoting conformational change, we run designs for pos allostery without the AntiMotifLoss. The results (linked Tab2) show that that the AntiMotifLoss is important for designing multistate behavior.
>
> With the extra space in the revision we are happy to use these results as a new benchmark for multistate design

---

### Official Review · Reviewer_AxE6 · 2026-03-13

**Soundness:** 3
**Presentation:** 3
**Significance:** 3
**Originality:** 3
**Overall Recommendation:** 4
**Confidence:** 4

**Summary:**

This paper proposes SWITCHCRAFT, which is a programmatic framework for generating proteins that activate, deactivate, or switch
between functional states. SWITCHCRAFT achieves backpropagation through compositional design constraints parameterized by structure prediction models. The paper evaluated on several design tasks and demonstrates the good performance of SWITCHCRAFT.

**Compliance With Llm Reviewing Policy:**

Affirmed.

**Key Questions For Authors:**

See above

**Limitations:**

See above

**Strengths And Weaknesses:**

# Strengths

1. The idea of developing a general-purpose computational framework for multistate protein design is very interesting and important for lots of real-world applications like enzyme design which has many transition states.

2. The paper is well written and easy to follow.

# Weaknesses

1. The hyperparameter values controlling different losses here seem largely influence the final model performance as stated in the four-stage optimization protocol. Did the author test the robustness of changing their values?

2. Enzymes have lots of transition states, and enzyme design is actually very suitable for the task the paper targets at. Can the author test the general enzyme design performance when binding to the specific substrates and execute the catalytic function?

---

> ### Author Rebuttal · Authors · 2026-03-25
>
> Thank you for the review and for appreciating the potential of our method! We address your questions and concerns below. We also have additional info/experiments for our responses to other reviewers **linked here for your reference (https://tinyurl.com/mrxy4utf).**
>
> ---
>
> ### **Question 1**
> > The hyperparameter values controlling different losses here seem largely influence the final model performance
>
> We want to clarify that there was no hyperparameter tuning across use cases. The same four stage optimization protocol is used throughout and is inherited directly from BoltzDesign1 (Cho et al 2025) and BindCraft (Pacesa et al, 2025) style methods. In that sense, we would not characterize SwitchCraft as relying on hyperparam tuning—we intend for it to be used out of the box for any use case a user would like to prescribe. We do agree that tuning could improve a particular design instance, but we intentionally avoided that here as it would counter the goal of our general programmatic workflow.
>
> ---
>
> ### **Question 2**
> > Can the author test the general enzyme design performance …  execute the catalytic function?
>
> Thank you for the suggestion; we completely agree that enzyme design is a natural application of multistate design methods. However, fully de novo enzyme design likely requires a design protocol capable of scaffolding _atomic motifs_ corresponding to catalytic sites. This is not yet possible with backprop-style methods such as SwitchCraft, which operate on token-level distograms output by co-folding models. Extending such methods to atomic motifs is the focus of our ongoing research. That being said, we have exciting news for SwitchCraft’s broader enzyme modeling capabilities (see below).
>
> ---
>
> ### **Wet lab validation for a SwitchCraft calcium modulated lysozyme**
>
> We are excited to see your interest in SwitchCraft applied to enzymes. Although not within the scope of this ICML submission, we are happy to report ongoing experimental validation of SwitchCraft in which we have engineered and experimentally verified a lysozyme that could be turned off by calcium. The protocol was largely unchanged (one state with calcium / one without) but an additional sequence similarity loss was to remain close to WT lysozyme and a motif loss was applied to restrict redesign to the catalytic residues. We are very excited to report that this yielded a functional lysozyme that showed activity in the absence of calcium and became inactive upon calcium binding. Our experimental results will be reported in a separate manuscript; however, we offer these results for your consideration of the merits of our computational manuscript. We hope they provide further evidence for the utility of the framework in general, and for enzymes in particular.
>
> ---
>
> We would like to thank the reviewer again for their comments, and we hope our clarifications together with the external motivating experimental results help address their concerns to justify raising the score!

---

> > ### Author Rebuttal · Reviewer_AxE6 · 2026-04-04
> >
> > Thanks for the author's response. It's good to know the wet-lab experiments on the lysozyme. I think it would definitely increase the quality of this paper if the authors put the experimental results on lysozyme after rebuttal.

---

> > > ### Author Response · Authors · 2026-04-04
> > >
> > > We are thrilled to hear that your concerns have been fully resolved! We are also glad to hear your enthusiasm about the lysozyme results, and also wish to highlight our other wet-lab experiments (see Fig1 of **https://tinyurl.com/mrxy4utf**). At your recommendation, we are happy to include some wet lab results in the revision. With the resolution of your concerns and the inclusion of wet-lab results, would you kindly consider increasing your score accordingly? Thank you!

---

### Official Review · Reviewer_28VJ · 2026-03-15

**Soundness:** 3
**Presentation:** 4
**Significance:** 4
**Originality:** 3
**Overall Recommendation:** 5
**Confidence:** 5

**Summary:**

This paper introduces a general framework for multistate protein design by bindcraft style sequence optimization through gradients of structure prediction models. This paper suggests many different design objective/constraints for different use cases, including positive/negative allostery and induced binding.

**Compliance With Llm Reviewing Policy:**

Affirmed.

**Final Justification:**

Rebuttal was strong and it reinforced my prior assessment of accept (5).

**Key Questions For Authors:**

1. I’m curious why the authors only test conformational changes under different ligand contexts. Many proteins naturally switch between multiple conformations even without any environmental change. So a setting with just ConfChangeLoss(stateA, stateB) plus appropriate constraints could be interesting. I'm curious if authors just didn't do it or if authors tried and didn't worked.

2. For the allostery examples, what is the baseline performance when enforcing only MotifLoss without AntiMotifLoss? Reporting this alongside the Appendix B success rate would help strengthen the paper’s claim that the more specific design specification is important.

**Limitations:**

I don't see the discussion on limitations and negative societal impact of their work. Please add this.

**Strengths And Weaknesses:**

### Strengths

- This paper tackles the very important problem of multistate protein design. The problem is very important with immense impact. To my knowledge, it is the first to approach this problem through a reasonably general computational framework. While I wouldn’t say the idea is entirely unprecedented, prior works have been quite problem-specific. Also, this paper discuss those prior works appropriately.

- The motivation and positioning of the paper are is strong. Conditional generation is unlikely to be effective with the absence of large-scale datasets with rich annotations of conformation-function relationships. This makes the proposed framework (construction of multiple structural constraints that describes function, and backpropagation from structure predictors) well justified.

- Includes many use cases (allostery, motif switching, ligand induced binding and more) with nice visualization!

- Very well written.

### Weaknesses
**Concerns and questions on binding loss.**

First of all, authors don’t state what H is and what does it mean to restrict to <20A. I understood as the entropy of the distogram probabilities restricted to bins <20A, after renormalizing those bins to sum to 1. If this understanding is correct, the paper should state it explicitly.

However, under that interpretation, the objective does not encourage making contacts, it only encourages the distogram distribution to be sharp within 20A. This is different from BindCraft loss, which encourage both contact formation and confidence in those contacts. If my interpretation is incorrect and the probabilities are not renormalized before computing the entropy, then the quantity appears somewhat ill-defined.

Also, I’m not fully convinced by the choice to optimize a distogram-based entropy objective instead of using confidence metrics derived from the PAE head, which is more standard in protein design works.

---

> ### Author Rebuttal · Authors · 2026-03-28
>
> We really appreciate your positive review and excitement for our paper! We hope to answer your questions below. We also have additional info/experiments for our responses to other reviewers **linked here for your reference (https://tinyurl.com/mrxy4utf).**
>
> ---
>
> ### **Contact and binding loss**
>
> For a given pair $i,j$, the contact / binding loss is
>
> $L_{ij} = -\mathbb{E}[\log P(D_{ij}) | D_{ij} < 20 A]$
>
> That is, it is the expectation of the log density (normalized over all distogram bins of $D_{ij}$) but taken only over the bins $<20 A$ (or 15 A for the binding loss). This can be more interpretably written as
>
> $L_{ij} = -\mathbb{E}[\log P(D_{ij} | D_{ij} < 20 A) | D_{ij} < 20 A] - \log P(D_{ij} < 20 A)$
>
> Namely, is is the sum of an well-defined entropy (encouraging sharp distograms $<20A$) and a contact probability. We will make sure to clarify this in the revision. We adopted this loss term without modification from BoltzDesign1 (Cho et al, 2025).
>
> ---
>
> ### **Distogram vs PAE loss**
>
> In diffusion-based co-folding  models (AF3-family), the confidence metrics are computed after long inference rollouts and are thus prohibitively expensive to optimize for design. We follow BoltzDesign1 (Cho et al, 2025), which uses the above distogram-based contact loss as a proxy for confidence. We agree that ideally, confidence metrics should be optimized directly; we are in the process of fine-tuning a confidence module that reads out directly from the pairformer output to make this feasible.
>
> ---
>
> ### **Conformational changes without environmental change**
>
> We thank you for this suggestion; indeed, many natural proteins do exhibit intrinsic ensembles. However, designing such ensembles would require a significantly different approach. In the absence of differing molecular contexts, the pairformer outputs for states A and B will be identical, making it impossible to optimize any measure of their difference. Instead, we would need to define new losses that extract and operate on latent representations of states _within_ disogram outputs. While this is thereotically possible, the degree to which co-folding models can meaningfully model intrinsic ensembles is unclear. Furthermore, **intrinsic ensembles do not provide interpretable state <-> function relationships**, so it is less clear what the immediate applications would be. For these reasons, we opted to focus on designing state switching that is regulated by environmental context.
>
> ---
>
> ### **Ablation experiment for positive allostery without AntiMotifLoss**
>
> Thank you for this suggestion, we have rerun positive allostery designs without the AntiMotifLoss. The results indicate that both the MotifLoss and AntiMotifLoss do seem necessary as removing the latter consistently reduces the number of successful designs.
>
> | Ligand | Motif | SwitchCraft | SwitchCraft No AntiMotifLoss |
> |---|---|---:|---:|
> | OQO | 3ixt | **5** | 1 |
> | OQO | 6e6r_long | **20** | 6 |
> | OQO | 6e6r_med | **15** | 5 |
> | OQO | 6e6r_short | **10** | 2 |
> | Zn²⁺ | 3ixt | **7** | 3 |
> | Zn²⁺ | 6e6r_long | **7** | 1 |
> | Zn²⁺ | 6e6r_med | **12** | 1 |
> | Zn²⁺ | 6e6r_short | 0 | 0 |
>
> ---
>
> ### **Wet-lab validation**
>
> Although not within the scope of this ICML submission, we are happy to report ongoing experimental validation of SwitchCraft in which we have designed and experimentally verified (1) a _de novo_ minibinder with function activated by an external analyte (https://tinyurl.com/mrxy4utf) , (2) allosteric deactivation of a wild-type enzyme (3) _de novo_ biosensor candidates that modulate fluorescence differently than WT GFP in the presence of the target analyte. Our experimental results will be reported in a separate manuscript; however, we offer these results for your consideration of the merits of our computational manuscript.
>
> ---

---

> > ### Author Rebuttal · Reviewer_28VJ · 2026-04-02
> >
> > Excited about new wet-lab results and thank you for doing the suggested ablation experiments.

---

> > > ### Author Response · Authors · 2026-04-07
> > >
> > > We are thrilled to hear that your concerns have been fully resolved! We are thankful for your comments, which have improved the presentation of our paper. We share your enthusiasm for the new wet lab results; if you feel this provides additional merit to the paper, we would greatly appreciate if you considered increasing the score!

---

### Decision · Program_Chairs · 2026-04-30

**Decision:**

Accept (regular)

**Comment:**

This paper introduces SwitchCraft, a programmatic framework for multistate protein design that formulates complex functional objectives as compositional constraints and optimizes sequences via backpropagation through structure prediction models. Reviewers broadly agree that the problem is important and underexplored, and find the framework conceptually compelling, well-motivated, and supported by diverse in silico demonstrations across multiple functional primitives. While concerns were raised regarding reliance on a single structure predictor, limited external validation, and the heuristic nature of the optimization pipeline, the rebuttal provides additional evidence—including independent model validation, ablations, and preliminary wet-lab results—that substantially strengthens confidence in the approach. Overall, the work represents a meaningful step toward programmable, higher-order protein design and opens a promising new direction for the field. I support acceptance.